# Meta-Reward-Net: Implicitly Differentiable Reward Learning for Preference-based Reinforcement Learning

**Runze Liu**[1,2]**, Fengshuo Bai**[3]**, Yali Du**[4,†]**, Yaodong Yang**[1,5,†]

[1]Institute for AI, Peking University, [2]Shandong University
[3]Institute of Automation, Chinese Academy of Science
[4]King's College London, [5]Beijing Institute for General AI
[†]: Corresponding to `yali.du@kcl.ac.uk`, `yaodong.yang@pku.edu.cn`

## Abstract

Setting up a well-designed reward function has been challenging for many reinforcement learning applications. Preference-based reinforcement learning (PbRL) provides a new framework that avoids reward engineering by leveraging human preferences (i.e., preferring apples over oranges) as the reward signal. Therefore, improving the efficacy of data usage for preference data becomes critical. In this work, we propose Meta-Reward-Net (MRN), a data-efficient PbRL framework that incorporates bi-level optimization for both reward and policy learning. The key idea of MRN is to adopt the performance of the Q-function as the learning target. Based on this, MRN learns the Q-function and the policy in the inner level while updating the reward function adaptively according to the performance of the Q-function on the preference data in the outer level. Our experiments on robotic simulated manipulation tasks and locomotion tasks demonstrate that MRN outperforms prior methods in the case of few preference labels and significantly improves data efficiency, achieving state-of-the-art in preference-based RL. Ablation studies further demonstrate that MRN learns a more accurate Q-function compared to prior work and shows obvious advantages when only a small amount of human feedback is available. The source code and videos of this project are released at `https://sites.google.com/view/meta-reward-net`[1].

## 1   Introduction

In recent years, reinforcement learning has achieved great success in solving complex sequential decision-making tasks, such as gaming AI [1, 2, 3, 4, 5], autonomous driving [6, 7], robotic manipulation [8, 9], order dispatching [10], population biology [11], quantitative finance [12], automation system [13, 14, 15], etc. For common decision making tasks, the goal of the agent is to maximize the cumulative reward. However, one central challenge to reinforcement learning is how to design reward functions. On the one hand, the quality of the designed reward function largely depends on the problem solver's understanding of task objective, operation logic, and related background knowledge. Even excellent engineers still need plenty of time to try different methods in complex RL tasks. On the other hand, there is a problem that the agent might hack the reward function. In policy learning, the agent utilizes the defect of the reward function to maximize the cumulative reward instead of solving the expected task. Besides, in human-involved scenarios, the objective of the agent is to maximize happiness of humans, making it hard to specify a reward function.

---

[1]Work done as a research intern at Peking University.

36th Conference on Neural Information Processing Systems (NeurIPS 2022).

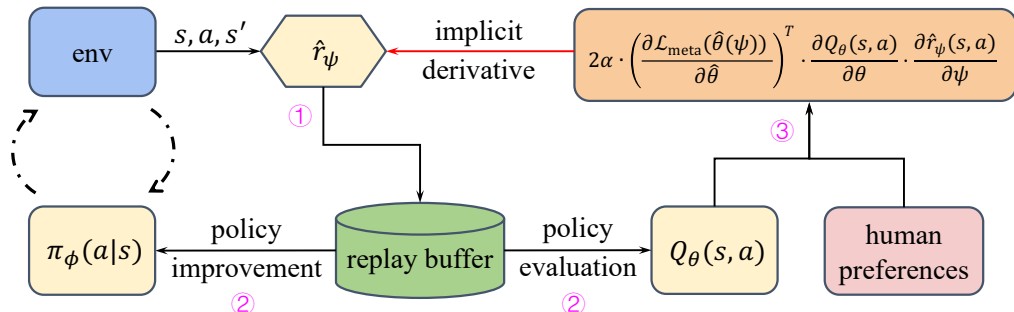

Figure 1: Framework of Meta-Reward-Net. ① Trajectories are sampled by interacting with the environment and reward is labeled by $\hat{r}_\psi$. ② Transitions are sampled from the replay buffer and are relabeled by the up-to-date $\hat{r}_\psi$ for optimizing the policy and the Q-function. ③ The performance of the Q-function on the preference data is evaluated to provide implicit derivative for reward learning.

Previous work has provided some ideas to avoid directly constructing reward functions, such as imitation learning [16]. Although imitation learning has an excellent performance in some tasks, its performance is difficult to surpass human level. Preference-based reinforcement learning is a more flexible and convenient alternative method. A human expert can easily give a preference to a trajectory pair of the agent, which implies the desired behaviors, that is to say, the goal of the task. In preference-based RL, the reward function is learned through preferences given by the human teacher on trajectory pairs. The feedback from humans guides the agent to achieve specified goals or learn desired behaviors. Recent work [17, 18, 19] shows that providing sufficient feedback achieves better performance under this setting.

However, preference data queried from human experts is expensive. In reality, there is likely to be only a small amount of data available. With the limitation of the amount of feedback, previous methods perform poorly or even do not work. Meanwhile, recent work on semi-supervised learning takes success in computer vision [20], pseudo labels can be adjusted according to the performance of the student and further improve student's performance. Inspired by this, we consider utilizing the performance from the Q-function on the preference labels for reward learning via bi-level optimization. By considering the implicit differentiation in reward learning, the reward function is aware of the accuracy of the Q-function, which is beneficial for the learning of the Q-function and the policy.

In this work, we focus on the efficiency of feedback in the learning of preference-based RL. We use bi-level optimization method for reward learning which contains two loops. In the inner loop, we update the Q-function and the policy via the reward function, while optimizing the reward function according to the performance of the Q-function on the preference data in the outer loop. Our experiments demonstrate that our method considerably improves feedback efficiency. Besides, the results of further evaluation show that MRN exceeds other methods by a large margin when few preference labels are available and learns a more accurate Q-function.

In summary, the main contributions of our work are three-fold. First, we propose a new preference-based RL algorithm Meta-Reward-Net, which utilizes bi-level optimization methods in reward learning. Second, we show that MRN substantially improves the feedback efficiency and outperforms preference-based RL baselines on a variety of robotic simulated manipulation tasks from Meta-world [21] and locomotion tasks from DeepMind Control Suite (DMControl) [22, 23]. Last, we demonstrate that benefiting from bi-level optimization, the advantage becomes obvious compared to PEBBLE when only few preference labels are provided. We also show that considering the performance of the Q-function in reward learning is beneficial for agent learning, leading to a more accurate Q-function and a better policy.

## 2 Related Work

**Reinforcement Learning.** Reinforcement learning has gradually become an effective and powerful method to solve complex sequential decision-making problems. In recent years, much prior work has proven the capacity of reinforcement learning. Reinforcement learning applications include

video games [1, 2, 3], robot control [24], manipulation [14, 15], board games [4], autonomous driving [6] and so on. In addition, there are many applications of reinforcement learning in the fields of computer vision [25], natural language processing [26], recommendation system [27, 28] and game theory [29, 30]. In the framework of reinforcement learning, an agent obtains data from interaction with the environment to optimize policy to maximize the expected return. In this process, the reward function plays a crucial role. Our method differs in that we do not assume that there is a reward function from engineering. Instead, we use the preferences of humans to guide the agent towards desired behaviors.

**Preference-based Reinforcement Learning.** Prior work has successfully trained the agent to complete specific tasks or achieve goals through the feedback from the teacher. [31] provides a general learning framework of preference-based reinforcement learning. [32] utilizes expert demonstrations and human preference feedback, initializes with imitation learning policy, and further improves the performance of the policy. In this work, we mainly focus on feedback efficiency in the learning of preference-based RL. Much previous work [33, 34, 35, 36, 37] considered learning reward function from the most informative data to be consistent with the human preferences. Recently, several feedback-efficient preference-based RL algorithms have been proposed. PEBBLE [17] combines unsupervised pre-training and the technique of relabeling experience to improve feedback efficiency. SURF [18] learns the reward function by semi-supervised learning and data augmentation. RUNE [19] facilitates exploration via reward uncertainty to reduce the number of preference labels. However, these methods only focus on guiding reward learning through supervised loss between human preferences and estimated preferences. We have a different approach in that in addition to utilizing the supervised loss, we also consider the performance of the Q-function on the labeled preference dataset to assist reward learning, thus beneficial for agent learning. By introducing this extra signal, MRN gets more task-related information from human preferences to improve feedback efficiency remarkably.

**Bi-level Optimization.** In computer vision, there are several bi-level optimization algorithms that achieve great success. Meta-Weight-Net [38] provides weights for samples from an unbalanced dataset in a bi-level manner, while Meta Label Correction [39] view this problem as a label correction problem. Meta Pseudo Labels [20] combines Pseudo Labels methods and bi-level optimization to generate high quality pseudo labels. In reinforcement learning, LIRPG [40] learns parametric intrinsic rewards and combines them with extrinsic rewards to improve the performance of policy-gradient-based learning methods. PR2 [41] and GR2 [42] propose to conduct multi-level recursive reasoning for multi-agent interactions. Bi-level method has also been studied for multi-agent coordination [43]. LIIR [44] learns an intrinsic reward function for each agent to achieve better cooperation in multi-agent reinforcement learning. CAIL [45] proposes to re-weight demonstrations of different optimality in imitation learning. [46] proposes a scalable meta-gradient framework to learn useful intrinsic reward functions to capture a rich form of knowledge across over multiple episodes. Neural auto-curricula [47] learns to create an autonomous and diverse agenda [48, 49] for solving two-player zero-sum games. Our method uses a similar bi-level method for reward learning. To the best of our knowledge, we are the first to introduce bi-level optimization into preference-based RL.

**Meta-learning from Demonstrations.** Previous research has many effects on meta-learning from demonstrations. [50] presents a new framework, PEMIRL, which enables meta-learning of rewards by leveraging unstructured demonstrations from multi-task. [51] presents a meta-learning method for one-shot imitation, which combine meta-learning with imitation learning. Specifically, this approach enables the agent to learn how to learn more efficiently and makes it acquire new skills from a single demonstration. [52] proposes a meta-learning algorithm that can utilize a small amount of trial experience to learn new behaviors illustrated by a single demonstration. [53] introduces an inverse reinforcement learning algorithm, which optimizes cost functions by differentiating at the inner optimization step. Unlike prior work, our method focuses on preference data rather than demonstrations, which is usually more expensive.

# 3 Preliminaries

**Preference-based Reinforcement Learning.** In standard reinforcement learning framework, a finite Markov decision process (MDP) can be presented as a tuple of $\langle \mathcal{S}, \mathcal{A}, R, P, \gamma \rangle$, which consists of state space $\mathcal{S}$, action space $\mathcal{A}$, transition function, reward function, and discount factor. $P(s'|s, a)$ represents stochastic dynamics of the environment, which is the probability of selecting action $a$ to

transit to $s'$ in a given state $s$. $R(s, a)$ represents the reward obtained by selecting an action $a$ in a given state $s$. The policy $\pi(a|s)$ is a mapping from state space to action space. The objective of the agent is to collect trajectories from interaction with the environment to maximize the expected return.

In the general preference-based RL from [31], there is no reward function from reward engineering and a reward function estimator $\hat{r}_\psi$ should be learned to be consistent with preferences from the human expert. Specifically, a segment $\sigma$ is a sequence of states and actions which is $(s_{t+1}, a_{t+1}, \cdots, s_{t+k}, a_{t+k})$. Human expert provides a preference $y$ on given two segments $(\sigma^0, \sigma^1)$ and $y$ is the distribution over $\{0, 1\}$, $y \in \{(1, 0), (0, 1), (0.5, 0.5)\}$. Following the Bradley-Terry model [54], a preference predictor constructed by the reward function estimate $\hat{r}_\psi$ is formulated as:

$$P_\psi[\sigma^0 \succ \sigma^1] = \frac{\exp \sum_t \hat{r}_\psi(s_t^0, a_t^0)}{\exp \sum_t \hat{r}_\psi(s_t^0, a_t^0) + \exp \sum_t \hat{r}_\psi(s_t^1, a_t^1)}, \tag{1}$$

where $\sigma^0 \succ \sigma^1$ denotes $\sigma^0$ is more consistent with the expectations of human experts compared with $\sigma^1$. The reward function learning can be solved by minimizing the cross-entropy loss between predictions from preference predictors and human preferences.

$$\mathcal{L}_{\text{supervised}}(\psi) = - \mathop{\mathbb{E}}_{(\sigma^0, \sigma^1, y) \sim \mathcal{D}} \Big[ y(0) \log P_\psi[\sigma^0 \succ \sigma^1] + y(1) \log P_\psi[\sigma^1 \succ \sigma^0] \Big]. \tag{2}$$

We refer to this objective as supervised loss in the following sections. By optimizing the reward function using this loss, segments that are more in line with human preferences obtain a higher cumulative reward.

**Soft Actor-Critic.** SAC [55] is an off-policy algorithm based on the maximum entropy RL, which encourages the agent to explore the environment by acting as randomly as possible. SAC consists of soft Q-function $Q_\theta(s, a)$ with parameters $\theta$ and policy $\pi_\phi(a|s)$ with parameters $\phi$. Q-function with parameters $\theta$ is defined as the expectation of return:

$$Q_\theta(s_t, a_t) = \mathbb{E} \left[ \sum_{t'=t}^{T} \gamma^{t'-t} r_{t'} \,\middle|\, S_t = s_t, A_t = a_t \right], \tag{3}$$

where $\gamma \in [0, 1]$ is a discount factor.

The parameters $\theta$ of soft Q-function are trained by minimizing the soft Bellman residual:

$$J_Q(\theta) = \mathbb{E}_{\tau_t \sim \mathcal{B}} \Big[ \big( Q_\theta(s_t, a_t) - r_t - \gamma \bar{V}(s_{t+1}) \big)^2 \Big], \tag{4}$$

where $\bar{V}(s_t) = \mathbb{E}_{a_t \sim \pi_\phi} \big[ Q_{\bar{\theta}}(s_t, a_t) - \alpha \log \pi_\phi(a_t|s_t) \big]$, $\tau_t = (s_t, a_t, s_{t+1}, r_t)$ is the transition at time step $t$, $\alpha$ is a learnable temperature parameter that controls the item of entropy, $\bar{\theta}$ are parameters of the target soft Q-function, and $\mathcal{B}$ is replay buffer. After the updating of the Q-function, policy $\pi_\phi$ is updated by minimizing the loss:

$$J_\pi(\phi) = \mathbb{E}_{s_t \sim \mathcal{B}, a_t \sim \pi_\phi} \Big[ \alpha \log \pi_\phi(a_t|s_t) - Q_\theta(s_t, a_t) \Big]. \tag{5}$$

By performing policy evaluation and policy improvement alternately, SAC trains an agent with excellent and stable performance. In this work, we consider using SAC as our backbone reinforcement learning algorithm.

# 4 Meta-Reward-Net

In this section, we formally present Meta-Reward-Net, which includes two key components, optimizing the reward function based on the performance of the Q-function in outer loop and learning the agent in the inner loop. In the following, we first provide a new perspective that the Q-function can be used to compute preference labels, then define the objective of MRN and formulate a bi-level optimization problem.

## 4.1 The Objective

The probability that segment $\sigma^0$ is preferred is proportional to the exponential return of it. Motivated by this, we use $Q_\theta(s_0^0, a_0^0)$ and $Q_\theta(s_0^1, a_0^1)$ to respectively measure of the return of $\sigma^0$ and $\sigma^1$ since

the Q-value equal to the expectation of segment return. Therefore, the probability that $\sigma^0$ is preferred to $\sigma^1$ is computed through the Q-function:

$$P_\theta[\sigma^0 \succ \sigma^1] = \frac{\exp Q_\theta(s_0^0, a_0^0)}{\exp Q_\theta(s_0^0, a_0^0) + \exp Q_\theta(s_0^1, a_0^1)}. \tag{6}$$

Given human preference labels $y$, the performance of the Q-function is measured by the cross-entropy loss between preference predictions computed by (6) and ground-truth labels:

$$\mathcal{L}_{\text{meta}}(\theta(\psi)) = - \mathop{\mathbb{E}}_{(\sigma^0, \sigma^1, y) \sim \mathcal{D}} \Big[ y(0) \log P_{\theta(\psi)}[\sigma^0 \succ \sigma^1] + y(1) \log P_{\theta(\psi)}[\sigma^1 \succ \sigma^0] \Big], \tag{7}$$

where $\theta(\psi)$ denotes the updating of $\theta$ depends on the reward provided by $\widehat{r}_\psi$.

The core of MRN is making reward learning be aware of the Q-function, which means that the optimization of the reward function takes the performance of current Q-function into consideration. Based on this idea, we formulate the objective using (7) as a measurement of the Q-function. The objective of MRN is to minimize the loss of $Q_\theta$ on a labeled preference dataset and the Q-function is trained by minimizing the Bellman residual. The overall objective is formulated as:

$$\begin{aligned} \min_{\psi, \theta} \quad & \mathcal{L}_{\text{meta}}(\theta(\psi)), \\ \text{s.t.} \quad & \theta(\psi) = \arg \min_\theta J_Q(\theta, \psi). \end{aligned} \tag{8}$$

By formulating MRN as a bi-level optimization algorithm, this allows the reward function to provide rewards that are beneficial for improving the performance of the Q-function, which further leads to a better policy $\pi_\phi$.

## 4.2 Bi-level Optimization

The objective in (8) is solved by the following bi-level optimization algorithm: $\theta$ is optimized by the reward estimation from $\widehat{r}_\psi$ in inner loop while $\psi$ is updated according to the performance of the Q-function on the labeled data in outer loop.

**Pseudo Updating: Building Connection between $\theta$ and $\psi$.** To improve the performance of Q-function, we formulate the outer loop updating as optimizing the loss of $\theta$ on the labeled preference data with respect to $\psi$. However, we can not directly optimize this objective since the objective is independent of $\psi$. So the first step is to build a connection between $\theta$ and $\psi$. Sample a mini-batch state-action pairs from the replay buffer and use them to query current reward function with parameters $\psi^{(k)}$ to obtain reward estimation $\widehat{r}_\psi(s_t, a_t)$, where $k$ denotes the current iteration step. Then (4) becomes:

$$J_Q(\theta) = \mathbb{E}_{\tau_t \sim \mathcal{B}} \Big[ \big( Q_\theta(s_t, a_t) - \widehat{r}_\psi(s_t, a_t) - \gamma \bar{V}(s_{t+1}) \big)^2 \Big]. \tag{9}$$

At each bi-level optimization step, we first pseudo update the parameters of the Q-function. Pseudo updating means that we do not directly perform the updating on the Q-function, but update the parameters of a copy of the current Q-function by minimizing (9) with learning rate $\alpha$:

$$\hat{\theta}^{(k)} = \theta^{(k)} - \alpha \left. \nabla_\theta J_Q(\theta, \psi) \right|_{\theta^{(k)}}, \tag{10}$$

where $\hat{\theta}^{(k)}$ denotes the updated copy of $\theta^{(k)}$. By performing (10), the connection between $\hat{\theta}^{(k)}$ and $\psi^{(k)}$ is built.

**Outer Loop: Optimizing $\psi$ to Improve the Performance of $Q_\theta$ on Labeled Data.** After building connections through pseudo updating, the copy of Q-function with parameters $\hat{\theta}^{(k)}$ is tested on labeled preference data. The predicted preference label $P_\theta(x)$ is computed by the Q-function using (6), where $x$ denotes a segment pair $(\sigma^0, \sigma^1)$. We use implicit differentiation in our method. The objective of outer loop is formulated in (7), and the implicit derivative of the outer loss with respect to $\psi$ is calculated using the chain rule:

$$g_{\text{meta}}^{(k)} = \left. \nabla_{\hat{\theta}} \mathcal{L}_{\text{meta}}(\hat{\theta}(\psi)) \right|_{\hat{\theta}^{(k)}} \left. \nabla_\psi \hat{\theta}^{(k)}(\psi) \right|_{\psi^{(k)}} = h \cdot \left. \nabla_\psi \widehat{r}(s_t, a_t; \psi) \right|_{\psi^{(k)}}, \tag{11}$$

where $h = 2\alpha \cdot \left( \nabla_{\hat{\theta}} \mathcal{L}_{\text{meta}}(\hat{\theta}^{(k)}) \right)^\top \cdot \nabla_\theta Q(s_t, a_t; \theta^{(k)})$ and full derivation can be found in Appendix B. (11) formulates the implicit derivative from the Q-function and this can be done easily using automatic

differentiation in Pytorch [56]. Since pseudo updating is performed and the connection between $\theta$ and $\psi$ is built, $\psi$ is updated to improve the performance of the Q-function by minimizing the cross-entropy between preference labels from $Q_\theta$ and ground-truth labels:

$$\psi^{(k+1)} = \psi^{(k)} - \beta \left. g_{\text{meta}}^{(k)} \right|_{\psi^{(k)}}, \tag{12}$$

where $\beta$ is the learning rate of the outer loop.

**Inner Loop: Optimizing $\theta$ and $\phi$.** In the inner loop, the objective is the same as (4) and (5) in the training of SAC. To calculate the new reward $\widehat{r}_\psi(s_t, a_t)$, use the same batch of state-action pairs in pseudo updating to query the updated reward function. However, we do not need the connection in the inner loop since the connection is used for outer level optimization. With newly obtained reward estimation, we update Q-function $Q_\theta$ by minimizing (4) with learning rate $\alpha$:

$$\theta^{(k+1)} = \theta^{(k)} - \alpha \left. \nabla_\theta J_Q(\theta) \right|_{\theta^{(k)}}, \tag{13}$$

and update policy $\pi_\phi$ by minimizing (5) with learning rate $\alpha$:

$$\phi^{(k+1)} = \phi^{(k)} - \alpha \left. \nabla_\theta J_\pi(\phi) \right|_{\phi^{(k)}}. \tag{14}$$

**Auxiliary Loss.** In addition to optimizing the loss in the outer loop, the reward function is augmented with a supervised loss generally used in preference-based RL, which is formulated in (2). Our intuition is that the optimization of the reward function needs both supervised learning and awareness of the performance of the Q-function, and neither of them can be removed. On the one hand, supervised loss is necessary for it ensures the reward estimation is consistent with human preferences. On the other hand, the outer loss is beneficial because it improves the performance of the Q-function, leading to a more accurate Q-function and finally a better policy. Additional experiments are conducted to discuss the relation between the two losses in Appendix F.

The full procedure of our method is detailed in Appendix A. Before reward learning, we first initialize the policy and replay buffer with unsupervised exploration, which is proposed in PEBBLE [17] and can be found in Appendix D. We use off-policy RL algorithm SAC to collect transitions and save them in the replay buffer. The Q-function and the policy is optimized in each training step.

### 4.3 Algorithm Convergence

For theoretical analysis, we provide convergence guarantee of Meta-Reward-Net. Theorem 1 demonstrates the convergence rate of the outer loss, while Theorem 2 shows the convergence of the inner loss. The detailed proofs of two theorems are presented in Appendix C.

**Theorem 1.** *Assume the outer loss $\mathcal{L}_{\text{meta}}$ is Lipschitz smooth with constant $L$, and the gradient of $\mathcal{L}_{\text{meta}}$ and $J_Q$ is bounded by $\rho$. Let $\widehat{r}_\psi$ be twice differential, with its gradient and Hessian respectively bounded by $\delta$ and $\mathcal{B}$. For some $c_1 > 0$, suppose the learning rate of the inner updating $\alpha_k = \min\{1, \frac{c_1}{T}\}$, where $c_1 < T$. For some $c_2 > 0$, suppose the learning rate of the outer updating $\beta_k = \min\{\frac{1}{L}, \frac{c_2}{\sqrt{T}}\}$, where $\frac{\sqrt{T}}{c_2} \geq L$, $\sum_{k=1}^{\infty} \beta_k \leq \infty$ and $\sum_{k=1}^{\infty} \beta_k^2 \leq \infty$. Meta-Reward-Net can achieve:*

$$\min_{1 \leq k \leq T} \mathbb{E}\left[ \left\| \nabla_\psi \mathcal{L}_{\text{meta}}(\hat{\theta}^{(k)}(\psi^{(k)})) \right\|^2 \right] \leq \mathcal{O}\left( \frac{1}{\sqrt{T}} \right). \tag{15}$$

**Theorem 2.** *Under the conditions in Theorem 1, Meta-Reward-Net can achieve:*

$$\lim_{k \to \infty} \mathbb{E}\left[ \left\| \nabla_\theta J_Q(\theta^{(k)}; \psi^{(k+1)}) \right\|^2 \right] = 0. \tag{16}$$

## 5 Experiments

In this section, our method is evaluated on a variety of robotic simulated manipulation tasks from Meta-world [21] and locomotion tasks from DeepMind Control Suite (DMControl) [22, 23]. The tasks used in our experiments are shown in Appendix E, which are the same as the tasks used in SURF.

## 5.1 Setup

**Baselines.** Reward-based SAC and three state-of-the-art preference-based RL algorithms are used for comparison:

- SAC [55]: SAC is considered as the upper bound algorithm since the agent is provided with ground-truth reward function, which is not the case in preference-based RL. SAC is evaluated in our experiments because it is the backbone RL algorithm of PEBBLE.

- Preference PPO [31]: the method is a re-implementation using PPO [57]. It uses an ensemble of reward functions and disagreement sampling for querying.

- PEBBLE [17]: the method is a preference-based RL method with unsupervised exploration and reward relabeling.

- SURF [18]: the method combines temporal data augmentation and pseudo labels in semi-supervised learning, which is the state-of-the-art algorithm in preference-based RL.

- Meta-Reward-Net (MRN): the proposed method, which is aware of the performance of the Q-function in reward learning through bi-level optimization.

**Implementation Details.** For all methods, we use unsupervised exploration proposed in PEBBLE [17]. For the sampling of queries, disagreement-based sampling is used for all preference-based RL methods, following the setting in [31]. An ensemble of three reward functions is used and the reward output is computed by averaging output of three reward functions. To systematically evaluate the performance and speed up the training process, following the setting in PEBBLE [17] and SURF [18], we consider a script teacher that can always provide the ground-truth preference label of a segment pair. Concretely, this is implemented by comparing the ground-truth return of each segment, but the reward is not accessible to the agent under the setting of preference-based RL.

For the implementation of SAC, Preference PPO and PEBBLE, we use the publicly released repository of B-Pref [58] in our experiments.[2] In their implementation, Preference PPO is re-implemented using on-policy algorithm PPO. SURF is also implemented using their released code.[3] For SAC, PEBBLE and SURF, the hyperparameters and network architectures we use are the same as them (e.g., number of network layers, learning rate, frequency of supervised reward learning). For the amount of human's preference feedback, we set 100 for Walker, Cheetah, Button Press and Window Open, 700 for Quadruped, 1000 for Door Open and Drawer Open, 4000 for Sweep Into, and 10000 for Hammer.

Our method is implemented by using PEBBLE as the backbone. We use bi-level updating frequency $N = 5000$ for Cheetah, Hammer, Button Press, Drawer Open and Window Open, $N = 1000$ for Walker, $N = 3000$ for Quadruped, and $N = 10000$ for Door Open and Sweep Into.

For each task, we run all algorithms independently for ten times and report the average with a standard deviation. Tasks of Meta-world are measured on success rate while the tasks of DMControl are measured on ground-truth episode return. The experiments are run on a single machine with one NVIDIA RTX 2080 Ti GPU. Details on hyperparameters, network architectures can be found in Appendix E.

## 5.2 Results

**Meta-world Tasks.** Examples of the six continuous control tasks from Meta-world are shown in Appendix E. These tasks are selected for our experiments, including robotic simulated manipulation skills of various difficulty. The details of the tasks can be found in Appendix E.

Figure 2 shows the training curves of MRN and the baselines on the Meta-world tasks. In this figure, SAC achieves the best performance in each task by using the ground-truth reward function. Since little human feedback is provided, we observe that there is a gap between all preference-based RL methods and the best performance, but MRN still exceeds the preference-based RL baselines by a large margin. These results demonstrate that MRN considerably improves the performance when only few preference labels are available. We also notice that Preference PPO does not work in most

---

[2] https://github.com/rll-research/BPref
[3] https://github.com/alinlab/SURF

tasks. The reason is that Preference PPO works well when the amount of preference labels is large. Once the number of labels reduces, the performance will have a significant drop.

**DMControl Tasks.** For DMControl, three locomotion tasks in Appendix E are used for evaluation, including Walker, Cheetah and Quadruped. These tasks encourage the agent to move forward by providing the agent with the reward that is positively correlated with agent's velocity. However, the agent is not accessible to the ground-truth reward function, and all preference-based RL methods are provided with human preference labels. Similar to tasks of Meta-world, we only provide the baselines and MRN with few but the same amount of preference labels.

Figure 3 shows the results of five methods on DMControl tasks. SURF achieves almost the same return as PEBBLE, while MRN shows obvious advantages compared with baseline methods. The results show that MRN performs well with few preference labels and improves feedback efficiency.

We remark that MRN can be regarded as being aware of the performance of the Q-function through bi-level optimization based on PEBBLE. Comparing the results of MRN and PEBBLE in Figure 2 and Figure 3, we find that using bi-level optimization can significantly improve performance when few preference labels are provided.

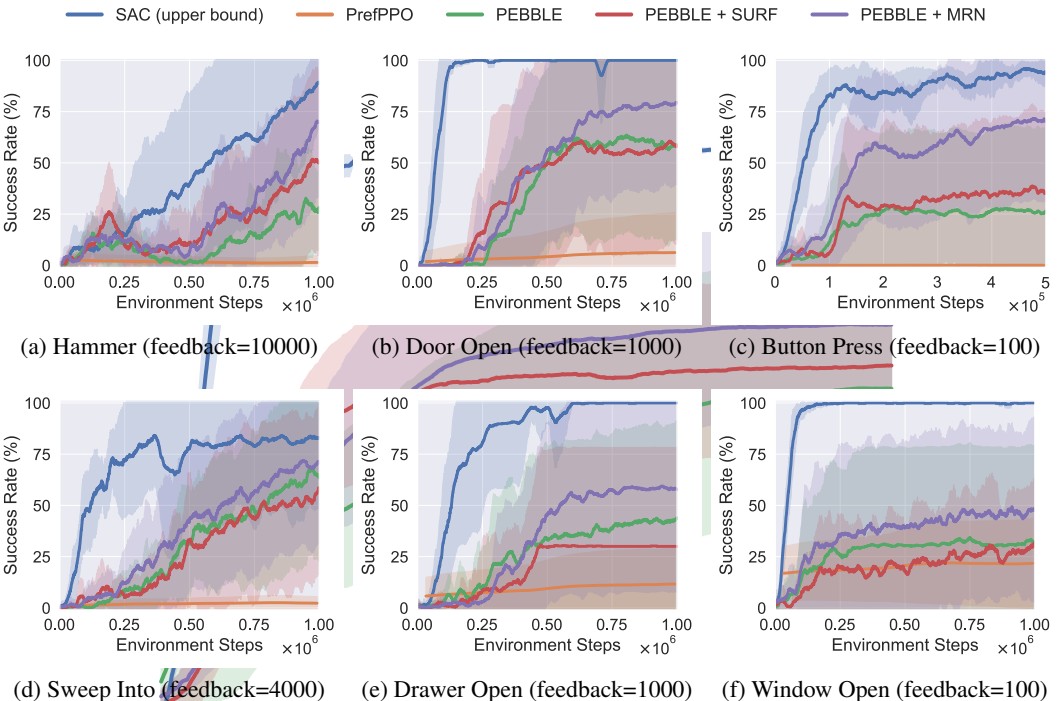

Figure 2: Training curves on six continuous control tasks from Meta-world. The solid line and shaded regions respectively denote mean and standard deviation of success rate, across ten runs.

### 5.3 Ablation Study

**Number of Human's Feedback.** Figure 2 and 3 show that our method outperforms Preference PPO, PEBBLE and SURF under relatively small amounts of human's feedback. To further analyze the performance of different algorithms with different number of preference labels, Extensive experiments are conducted on Walker and Door Open with varying amounts of feedback: $\{100, 400, 1000, 2000\}$. The results in Figure 4 suggest that our method performs better when the number of human's feedback is small. As the number increases, the performance of three algorithms becomes closer. Intuitively, this phenomenon could be considered that as feedback information becomes more sufficient, the performance gap caused by insufficient preference labels disappears.

**Accuracy of Q-function.** To compare the quality of Q-function trained by MRN and baselines, we use the mean squared error between ground-truth Q-values and the output of Q-function. Each method is evaluated on the same ten trajectories by calculating the MSE of the output of the Q-function

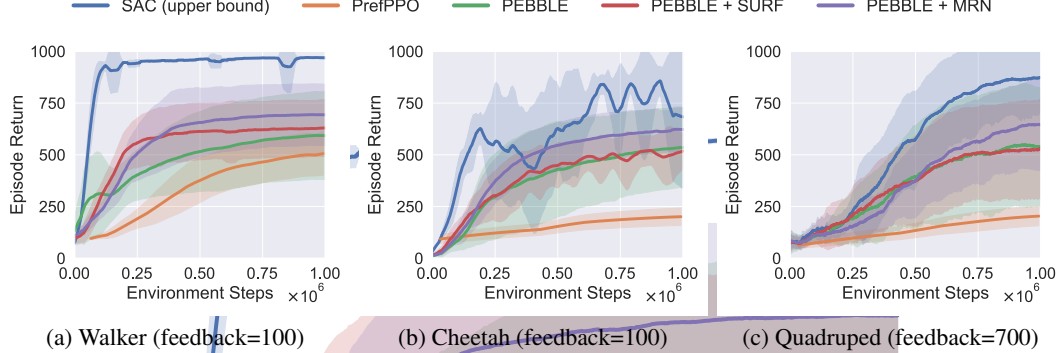

Figure 3: Training curves on three continuous control tasks from DMControl. The solid line and shaded regions respectively denote mean and standard deviation of success rate, across ten runs.

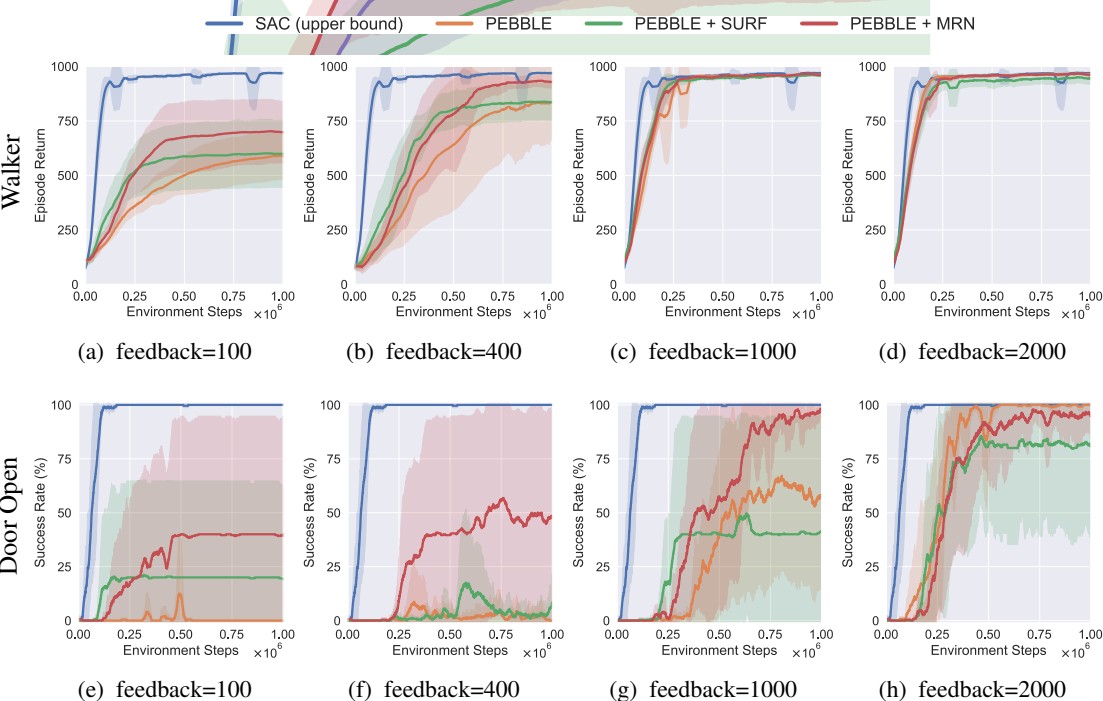

Figure 4: Training curves on Walker (first row) and Door Open (second row), measured by the ground-truth reward and success rate, respectively. The solid line and shaded regions respectively denote mean and standard deviation, across five runs.

and the ground-truth Q-values. We report mean and standard deviation across ten runs in Table 1. SAC is presented as the upper bound of the quality of Q-function and MRN achieves the lowest MSE among three methods. In MRN, reward learning additionally takes the performance of the Q-function into consideration. So the reward provided by the reward function is not only consistent with the ground-truth preference labels but also suitable for the performance of the current Q-function. Therefore, the Q-function learned by MRN is more accurate compared to the baselines.

Table 1: Mean squared error of learned Q-function across ten runs.

| Task/Method | PEBBLE | PEBBLE+SURF | PEBBLE+MRN | SAC (upper bound) |
|---|---|---|---|---|
| Walker | $0.12 \pm 0.04$ | $0.11 \pm 0.04$ | $\mathbf{0.10 \pm 0.02}$ | $0.07 \pm 0.02$ |
| Door Open | $0.52 \pm 0.05$ | $0.60 \pm 0.32$ | $\mathbf{0.48 \pm 0.06}$ | $0.45 \pm 0.01$ |

# 6   Conclusion

In this work, we propose Meta-Reward-Net, a novel feedback-efficient preference-based RL method. By incorporating bi-level optimization for reward and policy learning, we demonstrate MRN outperforms prior methods when a small amount of human feedback is available and considerably improves the feedback efficiency on a variety of robotic simulated tasks. In particular, our method exceeds the baselines by a large margin when there is few preference labels. From empirical results and analysis, we conclude that the efficiency of feedback in our method mainly benefits from: Firstly, our method learns a more accurate Q-function by aligning it with human preferences. Secondly, our method learns the Q-function and policy in the inner loop and optimizes reward function according to the performance of Q-function on the preference data in the outer loop. By this way, MRN successfully establishes an efficient mode of information transmission, which can extract more information. The potential negative social impact includes the carbon footprint of the experiments and future work based on MRN. We hope our method can provide inspiration for future work and encourage preference-based reinforcement learning to be better extended to practical applications.

**Limitations.** There are several limitations of MRN as follows. First, MRN relies on qualities of human feedback. Also, MRN cannot discriminate between good and bad actions within one trajectory. Last, representation learning on visual input requires additional parameters and thus requires more data. While this is not the focus of this work, we consider this as future works.

## Acknowledgements

Yaodong Yang is Sponsored by CAAI-Huawei MindSpore and CCF-Tencent Open Research Fund.

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
