# A  Meta-Reward-Net Algorithm

In this section, we present the detailed procedures of MRN in Algorithm 1. The updating of the reward function uses both the outer loss in (7) and supervised loss in (2). The reward function is updated using supervised loss per $K$ iterations, while the bi-level optimization is performed per $N$ iterations.

---

**Algorithm 1** Meta-Reward-Net

---

**Input:** supervised reward learning frequency $K$, bi-level updating frequency $N$
**Input:** number of human's preference labels per session $M$
1: Initialize $\theta$ and $\psi$
2: Initialize a preference dataset $\mathcal{D} \leftarrow \emptyset$
3: Initialize $\mathcal{B}$ and $\phi$ with unsupervised exploration
4: **for** each iteration **do**
5:     Take action $a_t \sim \pi_\phi(a_t|s_t)$ and obtain $s_{t+1} \sim p(s_{t+1} \mid s_t, a_t)$
6:     Store transition $\{(s_t, a_t, s_{t+1}, \widehat{r}_\psi(s_t, a_t))\}$ in $\mathcal{B}$
7:     Sample minibatch $\{(\tau_j)\}_{j=1}^{B} \sim \mathcal{B}$
8:     **if** iteration % $K$ == 0 **then**
9:         Query a human teacher for $M$ preference labels and store them in $\mathcal{D}$
10:        Sample preference data in $\mathcal{D}$
11:        Optimize (2) with respect to $\psi$
12:        Use updated $\widehat{r}_\psi$ to relabel the replay buffer $\mathcal{B}$
13:     **end if**
14:     **if** iteration % $N$ == 0 **then**
15:        Sample preference data in $\mathcal{D}$
16:        Pseudo update $\theta$ using (10)
17:        Update $\psi$ using (12)
18:        Use updated $\widehat{r}_\psi$ to relabel the replay buffer $\mathcal{B}$
19:     **end if**
20:     Update $\theta$ and $\phi$ using (13) and (14), respectively
21: **end for**
**Output:** policy $\pi_\phi$

---

# B  Derivation

In Section 4.2, the implicit derivative at iteration $k$ of $\psi$ is calculated by:

$$g_{\text{meta}}^{(k)} = \nabla_{\hat{\theta}} \mathcal{L}_{\text{meta}}(\hat{\theta}^{(k)}(\psi)) \nabla_\psi \hat{\theta}^{(k)}(\psi^{(k)}) = h \cdot \nabla_\psi \widehat{r}(s_t, a_t; \psi^{(k)}). \tag{17}$$

where $h$ is defined in (19). In this section, we present the full derivation of (17). Given trajectory pairs $x$ and their preference labels $y$, the parameters of the reward function $\psi$ is updated to minimize the cross-entropy loss between $y$ and the preference labels from the Q-function in the outer loop. Before the outer loop, pseudo updating is performed to build the connection between $\theta$ and $\psi$. In pseudo updating, a copy of the Q-function is updated to minimize the Bellman residual using the reward estimation from the reward function, which is formulated in (10). With the built connection, the gradient of $\psi$ with respect to $\mathcal{L}_{\text{meta}}$ is:

$$
\begin{aligned}
g_{\text{meta}}^{(k)} &= \left. \frac{\partial}{\partial \psi} \, \text{CE}(y, P_\theta(x; \hat{\theta}(\psi))) \right|_{\psi^{(k)}} \\
&= \left. \frac{\partial \text{CE}(y, P_\theta(x; \hat{\theta}(\psi)))}{\partial \hat{\theta}} \right|_{\hat{\theta}^{(k)}} \left. \frac{\partial \hat{\theta}^{(k)}(\psi)}{\partial \psi} \right|_{\psi^{(k)}} \\
&= \left. \frac{\partial \text{CE}(y, P_\theta(x; \hat{\theta}(\psi)))}{\partial \hat{\theta}} \right|_{\hat{\theta}^{(k)}} \left. \frac{\partial \hat{\theta}^{(k)}(\psi)}{\partial \widehat{r}(s_t, a_t; \psi)} \right|_{\psi^{(k)}} \left. \frac{\partial \widehat{r}(s_t, a_t; \psi)}{\partial \psi} \right|_{\psi^{(k)}} \\
&= 2\alpha \cdot \left. \frac{\partial \text{CE}(y, P_\theta(x; \hat{\theta}))}{\partial \hat{\theta}} \right|_{\hat{\theta}^{(k)}} \left. \frac{\partial Q(s_t, a_t; \theta)}{\partial \theta} \right|_{\theta^{(k)}} \left. \frac{\partial \widehat{r}(s_t, a_t; \psi)}{\partial \psi} \right|_{\psi^{(k)}},
\end{aligned}
\tag{18}
$$

where CE($\cdot$) denotes the cross-entropy loss. Let

$$h = 2\alpha \cdot \left( \frac{\partial \mathrm{CE}(y, P_\theta(x; \hat{\theta}))}{\partial \hat{\theta}} \bigg|_{\hat{\theta}^{(k)}} \right)^\top \cdot \frac{\partial Q(s_t, a_t; \theta)}{\partial \theta} \bigg|_{\theta^{(k)}}. \tag{19}$$

Then the parameters of the reward function are updated by (20):

$$\psi^{(k+1)} = \psi^{(k)} - \beta h \, \nabla_\psi \widehat{r}(s_t, a_t; \psi)|_{\psi^{(k)}}. \tag{20}$$

## C  Proofs for Algorithm Convergence

**Lemma 1.** *(Lemma 1.2.3 in [59]) If function $f(x)$ is Lipschitz smooth on $\mathbb{R}^n$ with constant L, then $\forall x_1, x_2 \in \mathbb{R}^n$, we have*

$$\left| f(x_2) - f(x_1) - f'(x_1)^\top (x_2 - x_1) \right| \leq \frac{L}{2} \|x_2 - x_1\|^2. \tag{21}$$

*Proof.* $\forall x_1, x_2 \in \mathbb{R}^n$, we have

$$
\begin{aligned}
f(x_2) &= f(x_1) + \int_0^1 f'(x_1 + \tau(x_2 - x_1))^\top (x_2 - x_1) d\tau \\
&= f(x_1) + f'(x_1)^\top (x_2 - x_1) + \int_0^1 [f'(x_1 + \tau(x_2 - x_1)) - f'(x_1)]^\top (x_2 - x_1) d\tau.
\end{aligned} \tag{22}
$$

Then we can derive that

$$
\begin{aligned}
\left| f(x_2) - f(x_1) - f'(x_1)^\top (x_2 - x_1) \right| &= \left| \int_0^1 [f'(x_1 + \tau(x_2 - x_1)) - f'(x_1)]^\top (x_2 - x_1) d\tau \right| \\
&\leq \int_0^1 \left| [f'(x_1 + \tau(x_2 - x_1)) - f'(x_1)]^\top (x_2 - x_1) \right| d\tau \\
&\leq \int_0^1 \|f'(x_1 + \tau(x_2 - x_1)) - f'(x_1)\| \cdot \|x_2 - x_1\| \, d\tau \\
&\leq \int_0^1 \tau L \|x_2 - x_1\|^2 \, d\tau = \frac{L}{2} \|x_2 - x_1\|^2,
\end{aligned} \tag{23}
$$

where the first inequality holds for $\left| \int_a^b f(x)dx \right| \leq \int_a^b |f(x)| \, dx$, the second inequality holds for Cauchy-Schwarz inequality, and the last inequality holds for the definition of Lipschitz smoothness. $\square$

**Lemma 2.** *Assume the outer loss $\mathcal{L}_{\mathrm{meta}}$ is Lipschitz smooth with constant L, and the gradient of the Q-function is bounded by $\rho$. Let $\widehat{r}_\psi$ be twice differential, with its gradient and Hessian respectively bounded by $\delta$ and $\mathcal{B}$. Then the gradient of $\psi$ with respect to the outer loss is Lipschitz continuous.*

*Proof.* The gradient of $\psi$ with respect to the outer loss is:

$$
\begin{aligned}
\nabla_\psi \mathcal{L}_{\mathrm{meta}}(\hat{\theta}^{(k)}(\psi)) \big|_{\psi^{(k)}} &= \nabla_{\hat{\theta}} \mathcal{L}_{\mathrm{meta}}(\hat{\theta}(\psi)) \big|_{\hat{\theta}^{(k)}} \nabla_\psi \hat{\theta}^{(k)}(\psi) \big|_{\psi^{(k)}} \\
&= 2\alpha \cdot \frac{\partial \mathcal{L}_{\mathrm{meta}}(\hat{\theta})}{\partial \hat{\theta}} \bigg|_{\hat{\theta}^{(k)}} \frac{\partial Q(s_t, a_t; \theta)}{\partial \theta} \bigg|_{\theta^{(k)}} \frac{\partial \widehat{r}(s_t, a_t; \psi)}{\partial \psi} \bigg|_{\psi^{(k)}} \\
&= h \cdot \frac{\partial \widehat{r}(s_t, a_t; \psi)}{\partial \psi} \bigg|_{\psi^{(k)}},
\end{aligned} \tag{24}
$$

where $h$ is defined in (19). Taking gradient of $\psi$ in both sides of (24), we obtain that

$$\nabla_{\psi^2}^2 \mathcal{L}_{\mathrm{meta}}(\hat{\theta}^{(k)}(\psi)) \big|_{\psi^{(k)}} = \frac{\partial h}{\partial \psi} \bigg|_{\psi^{(k)}} \frac{\partial \widehat{r}(s_t, a_t; \psi)}{\partial \psi} \bigg|_{\psi^{(k)}} + h \frac{\partial^2 \widehat{r}(s_t, a_t; \psi)}{\partial \psi^2} \bigg|_{\psi^{(k)}}. \tag{25}$$

The first term in (25) can be derived as follows:

$$
\begin{aligned}
\left\| \frac{\partial h}{\partial \psi}\Big|_{\psi^{(k)}} \frac{\partial \widehat{r}(s_t, a_t; \psi)}{\partial \psi}\Big|_{\psi^{(k)}} \right\| &\leq 2\alpha\delta \left\| \frac{\partial}{\partial \hat{\theta}}\left( \frac{\partial \mathcal{L}_{\text{meta}}(\hat{\theta})}{\partial \psi}\Big|_{\psi^{(k)}} \right)^{\top}\Big|_{\hat{\theta}^{(k)}} \frac{\partial Q(s_t, a_t; \theta)}{\partial \theta}\Big|_{\theta^{(k)}} \right\| \\
&= 4\alpha^2\delta \left\| \frac{\partial}{\partial \hat{\theta}}\left( \frac{\partial \mathcal{L}_{\text{meta}}(\hat{\theta})}{\partial \hat{\theta}}\Big|_{\hat{\theta}^{(k)}} \frac{\partial Q(s_t, a_t; \theta)}{\partial \theta}\Big|_{\theta^{(k)}} \frac{\partial \widehat{r}(s_t, a_t; \psi)}{\partial \psi}\Big|_{\psi^{(k)}} \right)^{\top}\Big|_{\hat{\theta}^{(k)}} \frac{\partial Q(s_t, a_t; \theta)}{\partial \theta}\Big|_{\theta^{(k)}} \right\| \\
&= 4\alpha^2\delta \left\| \left( \frac{\partial^2 \mathcal{L}_{\text{meta}}(\hat{\theta})}{\partial \hat{\theta}^2}\Big|_{\hat{\theta}^{(k)}} \frac{\partial Q(s_t, a_t; \theta)}{\partial \theta}\Big|_{\theta^{(k)}} \frac{\partial \widehat{r}(s_t, a_t; \psi)}{\partial \psi}\Big|_{\psi^{(k)}} \right)^{\top}\Big|_{\hat{\theta}^{(k)}} \frac{\partial Q(s_t, a_t; \theta)}{\partial \theta}\Big|_{\theta^{(k)}} \right\| \\
&\leq 4\alpha^2 L \rho^2 \delta^2,
\end{aligned}
\tag{26}
$$

since $\left\| \frac{\partial^2 \mathcal{L}_{\text{meta}}(\hat{\theta})}{\partial \hat{\theta}^2}\Big|_{\hat{\theta}^{(k)}} \right\| \leq L$, $\left\| \frac{\partial Q(s_t, a_t; \theta)}{\partial \theta}\Big|_{\theta^{(k)}} \right\| \leq \rho$, $\left\| \frac{\partial \widehat{r}(s_t, a_t; \psi)}{\partial \psi}\Big|_{\psi^{(k)}} \right\| \leq \delta$.

The second term in (25) can be derived as follows:

$$
\left\| h \frac{\partial^2 \widehat{r}(s_t, a_t; \psi)}{\partial \psi^2}\Big|_{\psi^{(k)}} \right\| = 2\alpha \left\| \frac{\partial \mathcal{L}_{\text{meta}}(\hat{\theta})}{\partial \hat{\theta}}\Big|_{\hat{\theta}^{(k)}}^{\top} \frac{\partial Q(s_t, a_t; \theta)}{\partial \theta}\Big|_{\theta^{(k)}} \frac{\partial^2 \widehat{r}(s_t, a_t; \psi)}{\partial \psi^2}\Big|_{\psi^{(k)}} \right\| \leq 2\alpha \mathcal{B} \rho^2,
\tag{27}
$$

since $\left\| \frac{\partial \mathcal{L}_{\text{meta}}(\hat{\theta})}{\partial \hat{\theta}}\Big|_{\hat{\theta}^{(k)}} \right\| \leq \rho$, $\left\| \frac{\partial^2 \widehat{r}(s_t, a_t; \psi)}{\partial \psi^2}\Big|_{\psi^{(k)}} \right\| \leq \mathcal{B}$.

Since $\|a + b\| \leq \|a\| + \|b\|$, we can derive that

$$
\begin{aligned}
\left\| \nabla^2_{\psi^2}\mathcal{L}_{\text{meta}}(\hat{\theta}^{(k)}(\psi))\Big|_{\psi^{(k)}} \right\| &= \left\| \frac{\partial h}{\partial \psi}\Big|_{\psi^{(k)}} \frac{\partial \widehat{r}(s_t, a_t; \psi)}{\partial \psi}\Big|_{\psi^{(k)}} + h \frac{\partial \widehat{r}(s_t, a_t; \psi)}{\partial \psi}\Big|_{\psi^{(k)}} \right\| \\
&\leq \left\| \frac{\partial h}{\partial \psi}\Big|_{\psi^{(k)}} \frac{\partial \widehat{r}(s_t, a_t; \psi)}{\partial \psi}\Big|_{\psi^{(k)}} \right\| + \left\| h \frac{\partial \widehat{r}(s_t, a_t; \psi)}{\partial \psi}\Big|_{\psi^{(k)}} \right\| \\
&\leq 2\alpha\rho^2(2\alpha L \delta^2 + \mathcal{B}).
\end{aligned}
\tag{28}
$$

According to Lagrange mean value theorem, $\forall \psi_1, \psi_2$, we have

$$
\left\| \nabla_{\psi}\mathcal{L}_{\text{meta}}(\hat{\theta}^{(k)}(\psi_1)) - \nabla_{\psi}\mathcal{L}_{\text{meta}}(\hat{\theta}^{(k)}(\psi_2)) \right\| \leq L' \|\psi_1 - \psi_2\|,
\tag{29}
$$

where $\nabla_{\psi}\mathcal{L}_{\text{meta}}(\hat{\theta}^{(k)}(\psi_1)) = \nabla_{\psi}\mathcal{L}_{\text{meta}}(\hat{\theta}^{(k)}(\psi))\Big|_{\psi_1}$, $L' = 2\alpha\rho^2(2\alpha L \delta^2 + \mathcal{B})$. $\qquad\square$

**Theorem 1.** *Assume the outer loss $\mathcal{L}_{\text{meta}}$ is Lipschitz smooth with constant $L$, and the gradient of $\mathcal{L}_{\text{meta}}$ and $J_Q$ is bounded by $\rho$. Let $\widehat{r}_{\psi}$ be twice differential, with its gradient and Hessian respectively bounded by $\delta$ and $\mathcal{B}$. For some $c_1 > 0$, suppose the learning rate of the inner updating $\alpha_k = \min\{1, \frac{c_1}{T}\}$, where $c_1 < T$. For some $c_2 > 0$, suppose the learning rate of the outer updating $\beta_k = \min\{\frac{1}{L}, \frac{c_2}{\sqrt{T}}\}$, where $\frac{\sqrt{T}}{c_2} \geq L$, $\sum_{k=1}^{\infty}\beta_k \leq \infty$ and $\sum_{k=1}^{\infty}\beta_k^2 \leq \infty$. Meta-Reward-Net can achieve:*

$$
\min_{1 \leq k \leq T} \mathbb{E}\left[ \left\| \nabla_{\psi}\mathcal{L}_{\text{meta}}(\hat{\theta}^{(k)}(\psi^{(k)})) \right\|^2 \right] \leq \mathcal{O}\left( \frac{1}{\sqrt{T}} \right).
\tag{30}
$$

*Proof.* The difference between $k+1$-th outer loss and $k$-th outer loss can be decomposed as:

$$
\begin{aligned}
&\mathcal{L}_{\text{meta}}(\hat{\theta}^{(k+1)}(\psi^{(k+1)})) - \mathcal{L}_{\text{meta}}(\hat{\theta}^{(k)}(\psi^{(k)})) \\
&= \left\{ \mathcal{L}_{\text{meta}}(\hat{\theta}^{(k+1)}(\psi^{(k+1)})) - \mathcal{L}_{\text{meta}}(\hat{\theta}^{(k)}(\psi^{(k+1)})) \right\} \\
&\quad + \left\{ \mathcal{L}_{\text{meta}}(\hat{\theta}^{(k)}(\psi^{(k+1)})) - \mathcal{L}_{\text{meta}}(\hat{\theta}^{(k)}(\psi^{(k)})) \right\}.
\end{aligned}
\tag{31}
$$

Since the outer loss function $\mathcal{L}_{\text{meta}}$ is Lipschitz smooth with constant $L$, the first term in (31) can be derived according to Lemma 1:

$$
\begin{aligned}
&\mathcal{L}_{\text{meta}}(\hat{\theta}^{(k+1)}(\psi^{(k+1)})) - \mathcal{L}_{\text{meta}}(\hat{\theta}^{(k)}(\psi^{(k+1)})) \\
&\leq \nabla_{\hat{\theta}}\mathcal{L}_{\text{meta}}(\hat{\theta}^{(k)}(\psi^{(k+1)}))^{\top}(\hat{\theta}^{(k+1)}(\psi^{(k+1)}) - \hat{\theta}^{(k)}(\psi^{(k+1)})) \\
&\quad + \frac{L}{2}\left\|\hat{\theta}^{(k+1)}(\psi^{(k+1)}) - \hat{\theta}^{(k)}(\psi^{(k+1)})\right\|^{2}.
\end{aligned}
\tag{32}
$$

For the first term in (32), since $\hat{\theta}^{(k+1)}(\psi^{(k+1)}) - \hat{\theta}^{(k)}(\psi^{(k+1)}) = -\alpha_k \nabla_\theta J_Q(\theta^{(k+1)})$, we have

$$
\begin{aligned}
&\nabla_{\hat{\theta}}\mathcal{L}_{\text{meta}}(\hat{\theta}^{(k)}(\psi^{(k+1)}))^{\top}(\hat{\theta}^{(k+1)}(\psi^{(k+1)}) - \hat{\theta}^{(k)}(\psi^{(k+1)})) \\
&\leq \left\|\nabla_{\hat{\theta}}\mathcal{L}_{\text{meta}}(\hat{\theta}^{(k)}(\psi^{(k+1)}))\right\| \cdot \left\|\hat{\theta}^{(k+1)}(\psi^{(k+1)}) - \hat{\theta}^{(k)}(\psi^{(k+1)})\right\| \\
&\leq \rho \cdot \left\|-\alpha_k \nabla_\theta J_Q(\theta^{(k+1)})\right\| \\
&\leq \alpha_k \rho^2,
\end{aligned}
\tag{33}
$$

where $\left\|\nabla_{\hat{\theta}}\mathcal{L}_{\text{meta}}(\hat{\theta}^{(k)}(\psi^{(k+1)}))\right\| \leq \rho$, $\left\|\nabla_\theta J_Q(\theta^{(k+1)})\right\| \leq \rho$ and the first inequality holds for Cauchy-Schwarz inequality. Combining (32) and (33), we obtain

$$
\mathcal{L}_{\text{meta}}(\hat{\theta}^{(k+1)}(\psi^{(k+1)})) - \mathcal{L}_{\text{meta}}(\hat{\theta}^{(k)}(\psi^{(k+1)})) \leq \alpha_k \rho^2 + \frac{L}{2}\alpha_k^2 \rho^2.
\tag{34}
$$

According to Lemma 2 and 1, the second term in (31) can be derived as follows:

$$
\begin{aligned}
&\mathcal{L}_{\text{meta}}(\hat{\theta}^{(k)}(\psi^{(k+1)})) - \mathcal{L}_{\text{meta}}(\hat{\theta}^{(k)}(\psi^{(k)})) \\
&\leq \nabla_{\psi}\mathcal{L}_{\text{meta}}(\hat{\theta}^{(k)}(\psi^{(k)}))^{\top}(\psi^{(k+1)} - \psi^{(k)}) + \frac{L}{2}\left\|\psi^{(k+1)} - \psi^{(k)}\right\|^{2} \\
&= -\beta_k \nabla_{\psi}\mathcal{L}_{\text{meta}}(\hat{\theta}^{(k)}(\psi^{(k)}))^{\top}\nabla_{\psi}\mathcal{L}_{\text{meta}}(\hat{\theta}^{(k)}(\psi^{(k)})) + \frac{L\beta_k^2}{2}\left\|\nabla_{\psi}\mathcal{L}_{\text{meta}}(\hat{\theta}^{(k)}(\psi^{(k)}))\right\|^{2} \\
&= -(\beta_k - \frac{L\beta_k^2}{2})\left\|\nabla_{\psi}\mathcal{L}_{\text{meta}}(\hat{\theta}^{(k)}(\psi^{(k)}))\right\|^{2},
\end{aligned}
\tag{35}
$$

since $\psi^{(k+1)} - \psi^{(k)} = -\beta_k \nabla_{\psi}\mathcal{L}_{\text{meta}}(\hat{\theta}^{(k)}(\psi^{(k)}))$. Combining (34) and (35), we have

$$
\begin{aligned}
&\mathcal{L}_{\text{meta}}(\hat{\theta}^{(k+1)}(\psi^{(k+1)})) - \mathcal{L}_{\text{meta}}(\hat{\theta}^{(k)}(\psi^{(k)})) \\
&\leq \alpha_k \rho^2 + \frac{L}{2}\alpha_k^2 \rho^2 - (\beta_k - \frac{L\beta_k^2}{2})\left\|\nabla_{\psi}\mathcal{L}_{\text{meta}}(\hat{\theta}^{(k)}(\psi^{(k)}))\right\|^{2}.
\end{aligned}
\tag{36}
$$

Summing up both sides in (36) from $k = 1$ to $T$, we obtain

$$
\begin{aligned}
&\sum_{k=1}^{T}(\beta_k - \frac{L\beta_k^2}{2})\left\|\nabla_{\psi}\mathcal{L}_{\text{meta}}(\hat{\theta}^{(k)}(\psi^{(k)}))\right\|^{2} \\
&\leq \mathcal{L}_{\text{meta}}(\hat{\theta}^{(1)}(\psi^{(1)})) - \mathcal{L}_{\text{meta}}(\hat{\theta}^{(T+1)}(\psi^{(T+1)})) + \sum_{k=1}^{T}(\alpha_k \rho^2 + \frac{L}{2}\alpha_k^2 \rho^2) \\
&\leq \mathcal{L}_{\text{meta}}(\hat{\theta}^{(1)}(\psi^{(1)})) + \sum_{k=1}^{T}(\alpha_k \rho^2 + \frac{L}{2}\alpha_k^2 \rho^2).
\end{aligned}
\tag{37}
$$

Therefore

$$
\min_{1 \le k \le T} \mathbb{E}\left[\left\|\nabla_\psi \mathcal{L}_{\text{meta}}(\hat{\theta}^{(k)}(\psi^{(k)}))\right\|^2\right]
$$

$$
\le \frac{\sum_{k=1}^{T}(\beta_k - \frac{L\beta_k^2}{2})\left\|\nabla_\psi \mathcal{L}_{\text{meta}}(\hat{\theta}^{(k)}(\psi^{(k)}))\right\|^2}{\sum_{k=1}^{T}(\beta_k - \frac{L\beta_k^2}{2})}
$$

$$
\le \frac{1}{\sum_{k=1}^{T}(2\beta_k - L\beta_k^2)}\left[2\mathcal{L}_{\text{meta}}(\hat{\theta}^{(1)}(\psi^{(1)})) + \sum_{k=1}^{T}(2\alpha_k\rho^2 + L\alpha_k^2\rho^2)\right]
$$

$$
\le \frac{1}{\sum_{k=1}^{T}\beta_k}\left[2\mathcal{L}_{\text{meta}}(\hat{\theta}^{(1)}(\psi^{(1)})) + \sum_{k=1}^{T}\alpha_k\rho^2(2 + L\alpha_k)\right]
$$

$$
\le \frac{1}{T\beta_k}\left[2\mathcal{L}_{\text{meta}}(\hat{\theta}^{(1)}(\psi^{(1)})) + T\alpha_1\rho^2(2 + L)\right] \tag{38}
$$

$$
= \frac{2\mathcal{L}_{\text{meta}}(\hat{\theta}^{(1)}(\psi^{(1)}))}{T}\frac{1}{\beta_k} + \frac{\alpha_1\rho^2(2 + L)}{\beta_k}
$$

$$
= \frac{2\mathcal{L}_{\text{meta}}(\hat{\theta}^{(1)}(\psi^{(1)}))}{T}\max\{L, \frac{\sqrt{T}}{c_2}\} + \min\{1, \frac{c_1}{T}\}\max\{L, \frac{\sqrt{T}}{c_2}\}\rho^2(2 + L)
$$

$$
\le \frac{2\mathcal{L}_{\text{meta}}(\hat{\theta}^{(1)}(\psi^{(1)}))}{c_2\sqrt{T}} + \frac{c_1\rho^2(2 + L)}{c_2\sqrt{T}}
$$

$$
= \mathcal{O}\left(\frac{1}{\sqrt{T}}\right),
$$

where the third inequality holds since $\sum_{k=1}^{T}(2\beta_k - L\beta_k^2) \ge \sum_{k=1}^{T}\beta_k$. $\qquad\square$

**Lemma 3.** *(Lemma A.5 in [60]) Let $(a_n)_{n\ge 1}, (b_n)_{n\ge 1}$ be two non-negative real sequences such that the series $\sum_{n=1}^{\infty}a_n$ diverges, the series $\sum_{n=1}^{\infty}a_nb_n$ converges, and there exists $K > 0$ such that $|b_{n+1} - b_n| \le Ka_n$. Then, the sequence $(b_n)_{n\ge 1}$ converges to 0.*

**Theorem 2.** *Assume the outer loss $\mathcal{L}_{\text{meta}}$ is Lipschitz smooth with constant $L$, and the gradient of $\mathcal{L}_{\text{meta}}$ and $J_Q$ is bounded by $\rho$. Let $\hat{r}_\psi$ be twice differential, with its gradient and Hessian respectively bounded by $\delta$ and $\mathcal{B}$. For some $c_1 > 0$, suppose the learning rate of the inner updating $\alpha_k = \min\{1, \frac{c_1}{T}\}$, where $c_1 < T$. For some $c_2 > 0$, suppose the learning rate of the outer updating $\beta_k = \min\{\frac{1}{L}, \frac{c_2}{\sqrt{T}}\}$, where $\frac{\sqrt{T}}{c_2} \ge L$, $\sum_{k=1}^{\infty}\beta_k \le \infty$ and $\sum_{k=1}^{\infty}\beta_k^2 \le \infty$. Meta-Reward-Net can achieve:*

$$
\lim_{k \to \infty} \mathbb{E}\left[\left\|\nabla_\theta J_Q(\theta^{(k)}; \psi^{(k+1)})\right\|^2\right] = 0. \tag{39}
$$

*Proof.* The difference between $k + 1$-th inner loss and $k$-th inner loss can be decomposed as:

$$
J_Q(\theta^{(k+1)}; \psi^{(k+2)}) - J_Q(\theta^{(k)}; \psi^{(k+1)})
$$
$$
= \left\{J_Q(\theta^{(k+1)}; \psi^{(k+2)}) - J_Q(\theta^{(k+1)}; \psi^{(k+1)})\right\} + \left\{J_Q(\theta^{(k+1)}; \psi^{(k+1)}) - J_Q(\theta^{(k)}; \psi^{(k+1)})\right\}. \tag{40}
$$

According to Lemma 1, the first term in (40) can be derived as follows:

$$
J_Q(\theta^{(k+1)}; \psi^{(k+2)}) - J_Q(\theta^{(k+1)}; \psi^{(k+1)})
$$
$$
\le \nabla_\psi J_Q(\theta^{(k+1)}; \psi^{(k+1)})^\top(\psi^{(k+2)} - \psi^{(k+1)}) + \frac{L}{2}\left\|\psi^{(k+2)} - \psi^{(k+1)}\right\|^2
$$
$$
= -\beta_{k+1}\nabla_\psi J_Q(\theta^{(k+1)}; \psi^{(k+1)})^\top\nabla_\psi \mathcal{L}_{\text{meta}}(\hat{\theta}^{(k+1)}(\psi^{(k+1)})) + \frac{L\beta_{k+1}^2}{2}\left\|\nabla_\psi \mathcal{L}_{\text{meta}}(\hat{\theta}^{(k+1)}(\psi^{(k+1)}))\right\|^2. \tag{41}
$$

And the second term in (40) can be derived as follows:

$$
\begin{aligned}
J_Q(\theta^{(k+1)}; \psi^{(k+1)}) - J_Q(\theta^{(k)}; \psi^{(k+1)}) &\le \nabla_\theta J_Q(\theta^{(k)}; \psi^{(k+1)})^\top (\theta^{(k+1)} - \theta^{(k)}) + \frac{L}{2} \left\| \theta^{(k+1)} - \theta^{(k)} \right\|^2 \\
&= -\alpha_k \nabla_\theta J_Q(\theta^{(k)}; \psi^{(k+1)})^\top \nabla_\theta J_Q(\theta^{(k)}; \psi^{(k+1)}) + \frac{L\alpha_k^2}{2} \left\| \nabla_\theta J_Q(\theta^{(k)}; \psi^{(k+1)}) \right\|^2 \\
&= -(\alpha_k - \frac{L\alpha_k^2}{2}) \left\| \nabla_\theta J_Q(\theta^{(k)}; \psi^{(k+1)}) \right\|^2 .
\end{aligned}
\tag{42}
$$

Combining (41) and (42), we obtain

$$
\begin{aligned}
&J_Q(\theta^{(k+1)}; \psi^{(k+2)}) - J_Q(\theta^{(k)}; \psi^{(k+1)}) \\
&\le -\beta_{k+1} \nabla_\psi J_Q(\theta^{(k+1)}; \psi^{(k+1)})^\top \nabla_\psi \mathcal{L}_{\text{meta}}(\hat{\theta}^{(k+1)}(\psi^{(k+1)})) + \frac{L\beta_{k+1}^2}{2} \left\| \nabla_\psi \mathcal{L}_{\text{meta}}(\hat{\theta}^{(k+1)}(\psi^{(k+1)})) \right\|^2 \\
&\quad - (\alpha_k - \frac{L\alpha_k^2}{2}) \left\| \nabla_\theta J_Q(\theta^{(k)}; \psi^{(k+1)}) \right\|^2 .
\end{aligned}
\tag{43}
$$

Taking expectation at both sides of (43), we can derive that

$$
\begin{aligned}
&\mathbb{E}\left[ J_Q(\theta^{(k+1)}; \psi^{(k+2)}) \right] - \mathbb{E}\left[ J_Q(\theta^{(k)}; \psi^{(k+1)}) \right] \\
&\le -\beta_{k+1} \mathbb{E}\left[ \left\| \nabla_\psi J_Q(\theta^{(k+1)}; \psi^{(k+1)}) \right\| \cdot \left\| \nabla_\psi \mathcal{L}_{\text{meta}}(\hat{\theta}^{(k+1)}(\psi^{(k+1)})) \right\| \right] \\
&\quad + \frac{L\beta_{k+1}^2}{2} \mathbb{E}\left[ \left\| \nabla_\psi \mathcal{L}_{\text{meta}}(\hat{\theta}^{(k+1)}(\psi^{(k+1)})) \right\|^2 \right] - (\alpha_k - \frac{L\alpha_k^2}{2}) \mathbb{E}\left[ \left\| \nabla_\theta J_Q(\theta^{(k)}; \psi^{(k+1)}) \right\|^2 \right] .
\end{aligned}
\tag{44}
$$

Rearranging the terms of (44) and summing up both sides of it from $k = 1$ to $\infty$, we have

$$
\begin{aligned}
&\sum_{k=1}^{\infty} \alpha_k \mathbb{E}\left[ \left\| \nabla_\theta J_Q(\theta^{(k)}; \psi^{(k+1)}) \right\|^2 \right] + \sum_{k=1}^{\infty} \beta_{k+1} \mathbb{E}\left[ \left\| \nabla_\psi J_Q(\theta^{(k+1)}; \psi^{(k+1)}) \right\| \cdot \left\| \nabla_\psi \mathcal{L}_{\text{meta}}(\hat{\theta}^{(k+1)}(\psi^{(k+1)})) \right\| \right] \\
&\le \sum_{k=1}^{\infty} \frac{L\alpha_k^2}{2} \mathbb{E}\left[ \left\| \nabla_\theta J_Q(\theta^{(k)}; \psi^{(k+1)}) \right\|^2 \right] + \mathbb{E}\left[ J_Q(\theta^{(1)}; \psi^{(2)}) \right] - \lim_{k\to\infty} \mathbb{E}\left[ J_Q(\theta^{(k+1)}; \psi^{(k+2)}) \right] \\
&\quad + \sum_{k=1}^{\infty} \frac{L\beta_{k+1}^2}{2} \mathbb{E}\left[ \left\| \nabla_\psi \mathcal{L}_{\text{meta}}(\hat{\theta}^{(k+1)}(\psi^{(k+1)})) \right\|^2 \right] \\
&\le \sum_{k=1}^{\infty} \frac{L(\alpha_k^2 + \beta_{k+1}^2)}{2} \rho^2 + \mathbb{E}\left[ J_Q(\theta^{(1)}; \psi^{(2)}) \right] \le \infty,
\end{aligned}
\tag{45}
$$

where the second inequality holds for $\sum_{k=1}^{\infty} \alpha_k^2 \le \infty$, $\sum_{k=1}^{\infty} \beta_k^2 \le \infty$, $\left\| \nabla_\theta J_Q(\theta^{(k)}; \psi^{(k+1)}) \right\| \le \rho$, $\left\| \nabla_\psi \mathcal{L}_{\text{meta}}(\hat{\theta}^{(k+1)}(\psi^{(k+1)})) \right\| \le \rho$.

Then we can derive that

$$
\begin{aligned}
&\left| \mathbb{E}\left[ \|\nabla_\theta J_Q(\theta^{(k+1)}; \psi^{(k+2)})\|^2 \right] - \mathbb{E}\left[ \|\nabla_\theta J_Q(\theta^{(k)}; \psi^{(k+1)})\|^2 \right] \right| \\
&= \left| \mathbb{E}\left[ (\|\nabla_\theta J_Q(\theta^{(k+1)}; \psi^{(k+2)})\| + \|\nabla_\theta J_Q(\theta^{(k)}; \psi^{(k+1)})\|) \cdot (\|\nabla_\theta J_Q(\theta^{(k+1)}; \psi^{(k+2)})\| - \|\nabla_\theta J_Q(\theta^{(k)}; \psi^{(k+1)})\|) \right] \right| \\
&\le \mathbb{E}\left[ \left| \|\nabla_\theta J_Q(\theta^{(k+1)}; \psi^{(k+2)})\| + \|\nabla_\theta J_Q(\theta^{(k)}; \psi^{(k+1)})\| \right| \left| \|\nabla_\theta J_Q(\theta^{(k+1)}; \psi^{(k+2)})\| - \|\nabla_\theta J_Q(\theta^{(k)}; \psi^{(k+1)})\| \right| \right] \\
&\le \mathbb{E}\left[ \left\| \nabla_\theta J_Q(\theta^{(k+1)}; \psi^{(k+2)}) + \nabla_\theta J_Q(\theta^{(k)}; \psi^{(k+1)}) \right\| \cdot \left\| \nabla_\theta J_Q(\theta^{(k+1)}; \psi^{(k+2)}) - \nabla_\theta J_Q(\theta^{(k)}; \psi^{(k+1)}) \right\| \right] \\
&\le \mathbb{E}\left[ (\left\| \nabla_\theta J_Q(\theta^{(k+1)}; \psi^{(k+2)}) \right\| + \left\| \nabla_\theta J_Q(\theta^{(k)}; \psi^{(k+1)}) \right\|) \cdot \left\| \nabla_\theta J_Q(\theta^{(k+1)}; \psi^{(k+2)}) - \nabla_\theta J_Q(\theta^{(k)}; \psi^{(k+1)}) \right\| \right] \\
&\le 2L\rho \mathbb{E}\left[ \left\| (\theta^{(k+1)}, \psi^{(k+2)}) - (\theta^{(k)}, \psi^{(k+1)}) \right\| \right] \\
&\le 2L\rho \alpha_k \beta_{k+1} \mathbb{E}\left[ \left\| \left( \nabla_\theta J_Q(\theta^{(k)}; \psi^{(k+1)}), \nabla_\psi \mathcal{L}_{\text{meta}}(\hat{\theta}^{(k+1)}(\psi^{(k+1)})) \right) \right\| \right] \\
&\le 2L\rho \alpha_k \beta_{k+1} \sqrt{ \mathbb{E}\left[ \left\| \nabla_\theta J_Q(\theta^{(k)}; \psi^{(k+1)}) \right\|^2 \right] + \mathbb{E}\left[ \left\| \nabla_\psi \mathcal{L}_{\text{meta}}(\hat{\theta}^{(k+1)}(\psi^{(k+1)})) \right\|^2 \right] } \\
&\le 2L\rho \alpha_k \beta_{k+1} \sqrt{2\rho^2} \\
&\le 2\sqrt{2} L\rho^2 \beta_1 \alpha_k,
\end{aligned}
\tag{46}
$$

where the second inequality holds for $|(\|a\| + \|b\|)(\|a\| - \|b\|)| \le \|a + b\| \|a - b\|$.

Since

$$\sum_{k=1}^{\infty} \beta_{k+1} \mathbb{E}\left[\left\|\nabla_\psi J_Q(\theta^{(k+1)}; \psi^{(k+1)})\right\| \cdot \left\|\nabla_\psi \mathcal{L}_{\text{meta}}(\hat{\theta}^{(k+1)}(\psi^{(k+1)}))\right\|\right] \le L\rho \sum_{k=1}^{\infty} \beta_{k+1} \le \infty,$$

(47)

according to (45), we have

$$\sum_{k=1}^{\infty} \alpha_k \mathbb{E}\left[\left\|\nabla_\theta J_Q(\theta^{(k)}; \psi^{(k+1)})\right\|^2\right] < \infty$$

(48)

Since $\sum_{k=1}^{\infty} \alpha_k = \infty$, according to Lemma 3, we have

$$\lim_{k\to\infty} \mathbb{E}\left[\left\|\nabla_\theta J_Q(\theta^{(k)}; \psi^{(k+1)})\right\|^2\right] = 0.$$

(49)

$\square$

## D   Details of PEBBLE

In this section, we present details of unsupervised exploration and sampling scheme used in our experiments in Section 5.1. These two techniques play an essential role in improving the feedback efficiency of PEBBLE [17]. In our experiments, all preference-based RL algorithms use unsupervised exploration and disagreement-based sampling for fairness.

### D.1   Unsupervised Pre-training

For a random policy, in the initial stage of training, the generated trajectories are difficult to cover the trajectory space, or the agent lacks behavioural diversity. As a result, the information conveyed by human preference feedback is limited. Even worse, it might be difficult for human experts to give preferences to trajectory pairs (e.g., a pair of poor trajectories.). This problem leads to a significant impact on the efficiency of the feedback in the initial stage. A practical solution is to encourage the agent to explore the environment and improve the diversity of trajectories. For specific implementation, the entropy of the agent's state can be considered:

$$\widehat{\mathcal{H}}(s) = -\mathbb{E}_{s\sim p(s)}\left[\log p(s)\right] \propto \frac{1}{N} \sum_i^N \log(\|s_i - s_i^k\|),$$

(50)

where $\widehat{\mathcal{H}}$ is the particle-based entropy estimator and $s_i^k$ denotes the $k$-th nearest neighbor of $s_i$. Specifically, the intrinsic reward is constructed to facilitate the agent to explore the environment and cover a broader range of state space. By maximizing the accumulation of intrinsic reward, the agent is effectively encouraged to explore the environment and improve the diversity of agent trajectories. Inspired by this, PEBBLE [17] proposed a method to construct intrinsic reward by using state entropy. It defines the intrinsic reward as the formula (51).

$$r^{\text{int}}(s_t) = \log(\|s_t - s_t^k\|).$$

(51)

(51) represents the immediate reward of $s_t$ depends on the distance between $s_t$ and the $k$-th nearest neighbor of $s_t$. In our experiment, to ensure the fairness of the experiment, all algorithms use the technique of unsupervised exploration.

### D.2   Disagreement Sampling

The sampling scheme is a strategy for selecting queries, which will have an impact on feedback efficiency during reward learning. Disagreement sampling [31] is one of the sampling schemes used in this work, the same as PEBBLE [17] and SURF [18]. It selects segments from the human preference dataset with high variance across an ensemble of reward models. Intuitively, the variance of the predicted return of segment across multiple reward models approximates the reward models' uncertainty to a certain extent.

# E  Experimental Details

## E.1  Tasks

The locomotion tasks from DeepMind Control Suite (DMControl) [22, 23] and simulated robotic manipulation tasks from Meta-world [21] used in our experiments are shown in Figure 5.

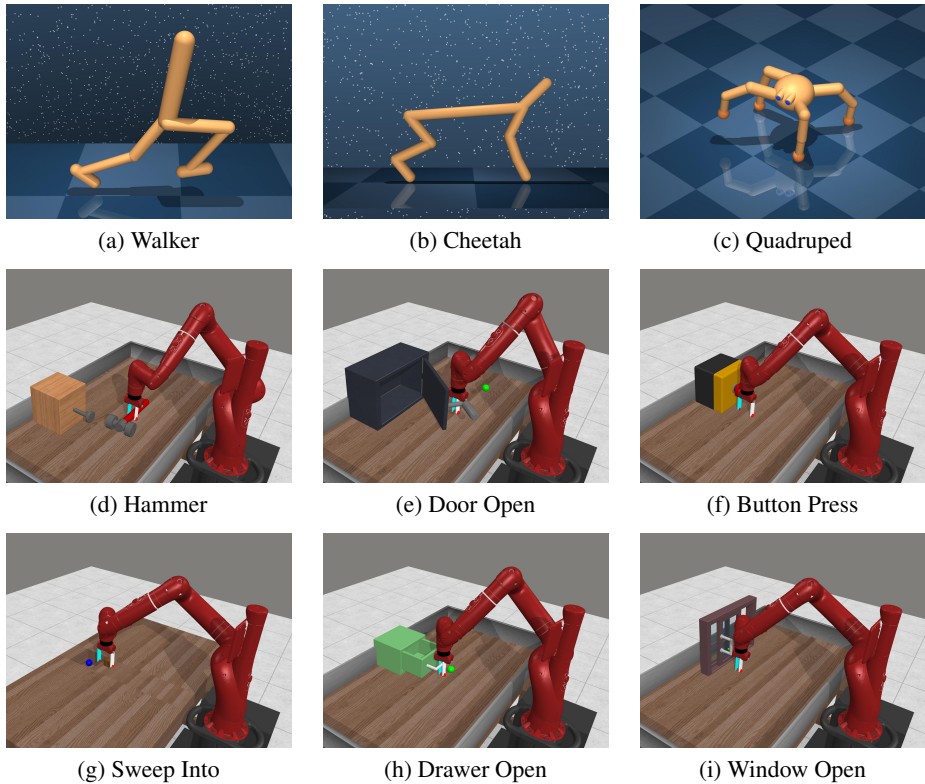

Figure 5: Nine tasks used for experiments. (a-c) DMControl tasks. (d-i) Meta-world tasks.

**DMControl Tasks.**

- Walker: A planar walker learns to control its torso and walk on the ground.
- Cheetah: A planar biped learns to control its body and run on the ground.
- Quadruped: A quadruped ant learns to control its body and limbs and crawl on the ground.

**Meta-world Tasks.**

- Hammer: An agent controls the robotic arm to hammer a screw into the wall. The initial positions of the hammer and screw are random.
- Door Open: An agent controls the robotic arm to open a door with a revolving joint. The initial position of the door is random.
- Button Press: An agent controls the robotic arm to press a button. The initial position of the button is random.
- Sweep Into: An agent controls the robotic arm to sweep a ball into the hole. The initial position of the ball is random.
- Drawer Open: An agent controls the robotic arm to open a drawer. The initial position of the drawer is random.
- Window Open: An agent controls the robotic arm to open a window. The initial position of the window is random.

### E.2 Implementation Details

For SAC, the agent is provided the ground-truth reward function and SAC serves as the upper bound of all methods. The detailed hyperparameters of SAC are shown in Table 2. For Preference PPO, we follow the hyperparameters settings used in PEBBLE [17], which are shown in Table 3. The settings of PEBBLE are kept the same with their implementation, and they are detailed in Table 4. For SURF, most hyperparameters are the same as those of PEBBLE in Table 4 and other hyperparameters are detailed in Table 5. The reward model is an ensemble of three MLPs. Each MLP has three layers with 256 hidden units. The output of the reward model is limited within $[-1, 1]$ using tanh activation. The hyperparameters of MRN are the same as those of PEBBLE in Table 4. The only difference is the bi-level updating frequency, which is detailed in Section 5.1.

Table 2: Hyperparameters of SAC.

| Hyperparameter | Value | Hyperparameter | Value |
|---|---|---|---|
| Number of layers | 2 (DMControl), 3 (Meta-world) | Initial temperature | 0.1 |
| Hidden units of each layer | 1024 (DMControl), 256 (Meta-world) | Optimizer | Adam |
| Learning rate | 0.0005 (Walker), 0.001 (Cheetah) | Critic target update freq | 2 |
| | 0.0001 (Quadruped), 0.0003 (Meta-world) | Critic EMA $\tau$ | 0.005 |
| Batch size | 1024 (DMControl), 512 (Meta-world) | $(\beta_1, \beta_2)$ | $(0.9, 0.999)$ |
| Steps of unsupervised pre-training | 9000 | Discount $\gamma$ | 0.99 |

Table 3: Hyperparameters of Preference PPO.

| Hyperparameter | Value | Hyperparameter | Value |
|---|---|---|---|
| Learning rate | 0.0003 (Meta-world), 0.0001 (Cheetah) | Batch Size | 512 (Cheetah) |
| | $5e^{-5}$ (Walker, Quadruped) | | 128 (others) |
| Number of parallel environments | 32 (Walker, Meta-world) | GAE parameter $\lambda$ | 0.92 |
| | 16 (Cheetah, Quadruped) | Entropy coefficient | 0.0 |
| Steps of unsupervised pre-training | 32000 | Discount $\gamma$ | 0.99 |

Table 4: Hyperparameters of PEBBLE.

| Hyperparameter | Value |
|---|---|
| Length of segment | 50 |
| Learning rate | 0.0005 (Walker, Cheetah), 0.0001 (Quadruped), 0.0003 (Meta-world) |
| Frequency of feedback | 20000 (Walker, Cheetah), 30000 (Quadruped), 5000 (Meta-world) |
| Amount of feedback / | 700/35 (Quadruped), 100/10 (Walker, Cheetah) |
| feedback amount per session | 10000/50 (Hammer), 4000/20 (Sweep Into) |
| | 1000/10 (Door Open, Drawer Open), 100/5 (Button Press, Window Open) |
| Number of reward functions | 3 |

Table 5: Hyperparameters of SURF.

| Hyperparameter | Value |
|---|---|
| Unlabeled batch ratio $\mu$ | 4 |
| Threshold $\tau$ | 0.999 (Cheetah, Window Open, Sweep Into) |
| | 0.99 (others) |
| Loss weight $\lambda$ | 1 |
| Min/Max length of cropped segment $[H_{\mathtt{min}}, H_{\mathtt{max}}]$ | $[45, 55]$ |
| Segment length before cropping | 60 |

## F Additional Results

To further study the properties of MRN, we additionally conduct a range of supplementary experiments which include the impact of bi-level updating frequency and supervised reward learning frequency on the performance of MRN, and the robustness of MRN with noisy preference labels.

**Bi-level Updating Frequency.** To investigate how bi-level updating frequency influences our method, we conduct several experiments of different bi-level updating frequency settings. For Walker, MRN is evaluated using $N \in \{400, 800, 1000, 1200, 1600\}$. MRN is evaluated using $N \in \{1000, 5000, 10000, 15000, 20000\}$ for Door Open and $N \in \{1000, 3000, 5000, 7500, 10000\}$ for Hammer and Window Open. It is shown in Figure 6 that under the same amount of human's preference labels, our method performs well when the bi-level updating frequency is moderate, i.e., neither too low or too high. When the frequency is too high (i.e., small $N$), the inner level is not fully optimized. When the frequency is too low (i.e., large $N$), bi-level optimization has a relatively small influence on the performance of MRN.

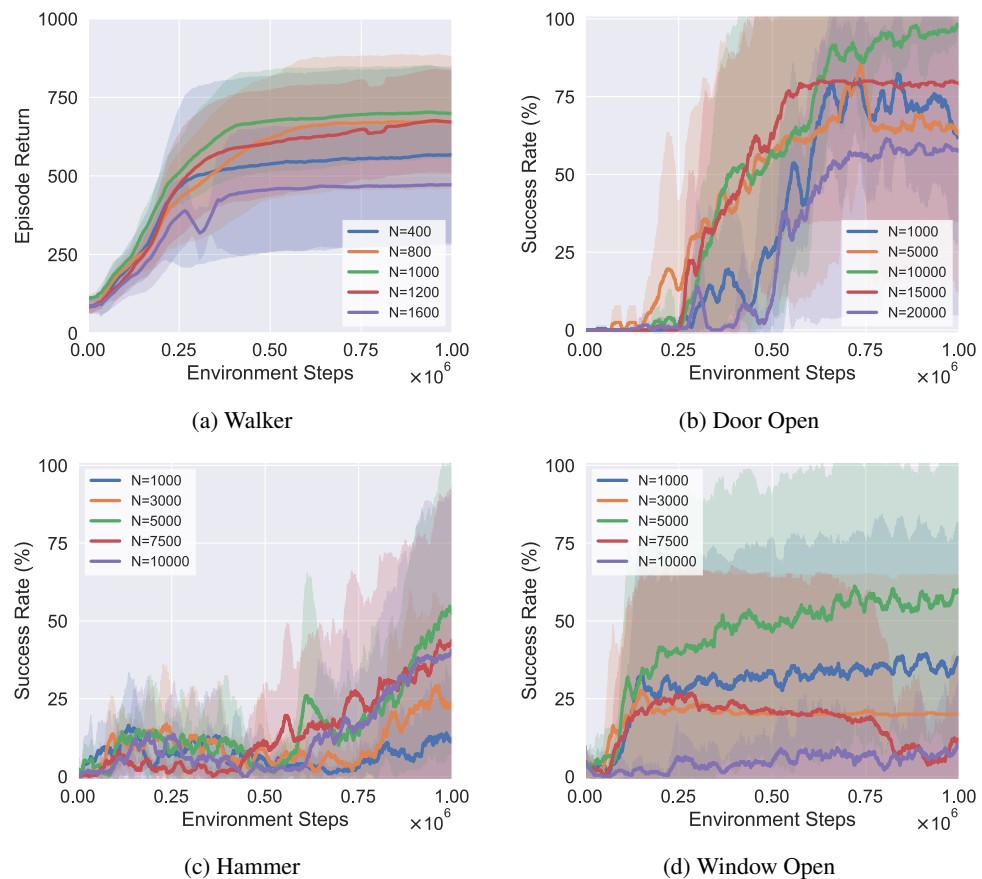

Figure 6: Training curves on Walker, Door open, Hammer and Window Open with different bi-level updating frequencies. The solid line and shaded regions respectively denote mean and standard deviation, across five runs.

**Supervised Loss Updating Frequency.** To investigate how supervised loss influences our method, we evaluate MRN using different supervised loss updating frequency $K$. $K \in \{10000, 20000, 30000, 40000\}$ are used for Walker and $K \in \{1000, 3000, 5000, 7500, 10000\}$ are used for Door Open. Figure 7 demonstrates that MRN learns both quickly and well when $K$ is moderate. When $K$ is small, the influence of supervised loss exceeds that of bi-level updating loss, and the performance is close to that of PEBBLE. When $K$ goes larger, MRN learns slower at initial stage, but finally learns a good policy.

**Robustness of MRN.** To evaluate the robustness of our method, we conduct extensive experiments when 10% of human preference data is modified at random (i.e., when the label is $(1, 0)$, it will be changed to $(0, 1)$, and vice versa). In this way, introducing erroneous data simulates that human experts make mistakes with a constant probability when giving preferences to trajectory pairs. Figure 8 shows that MRN has excellent robustness with 10% erroneous data. When there are a few errors in the preferences, the performance of all methods are affected. However, MRN is more robust

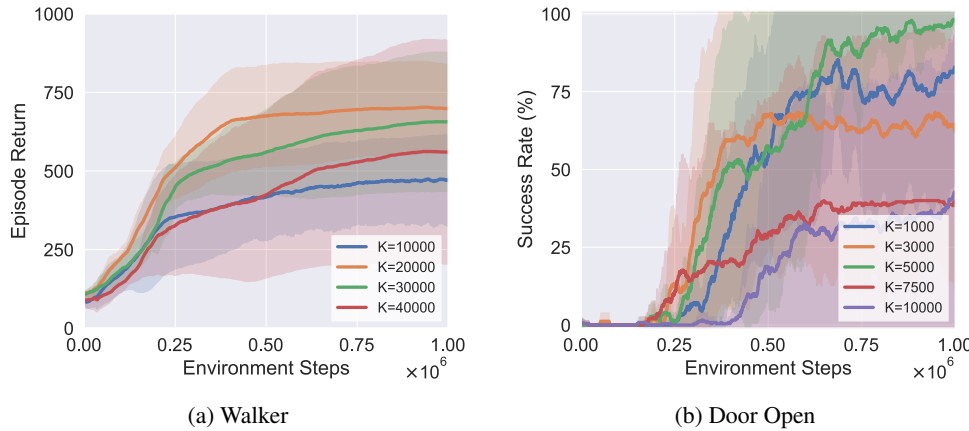

Figure 7: Training curves on Walker and Door Open with different supervised loss updating frequencies. The solid line and shaded regions respectively denote mean and standard deviation, across five runs.

and outperforms PEBBLE and SURF with noisy labels. We remark that the performance of MRN with noisy preferences is the same as that of PEBBLE trained by perfect data on Walker. Likewise, the performance of MRN with noisy labels is the same as that of PEBBLE trained by perfect data on Door Open.

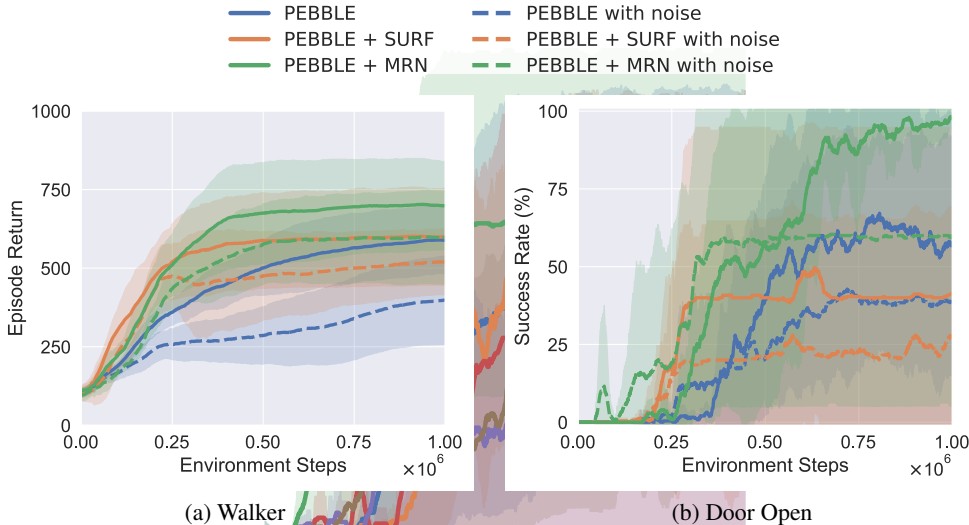

Figure 8: Training curves on Walker and Door Open. The solid/dashed line and shaded regions respectively denote mean and standard deviation, across five runs.