# OpenReview forum: "Meta-Reward-Net: Implicitly Differentiable Reward Learning for Preference-based Reinforcement Learning"
_NeurIPS.cc/2022/Conference — NeurIPS 2022 Accept_

### Official Review · Reviewer_gArn · 2022-07-10

**Rating:** 7
**Confidence:** 4
**Soundness:** 3 good
**Presentation:** 4 excellent
**Contribution:** 3 good

**Summary:**

This work introduces a meta-learning based interpretation of preference-based reinforcement learning. The proposed method MRN utilises bi-level optimisation to learn reward functions using expert preferences, a method introduced in [23]. The novelty of this work is in the implicit conditioning of the reward learning process on the values obtained from the Q function. The work is evaluated thoroughly in simulation and compared to reasonable baselines. The discussed results show the faster and more efficient (in terms of required expert labels) learning process of MRN. Overall, my opinion of this work is generally positive.

However, I do have a few concerns and suggestions. The strongest reservations I have are that there is **no sufficient motivation** for using the proposed approach over alternative meta-learning based reward learning solutions. Furthermore, at this point it seems like the proposed solution has a **fairly moderate impact** (more details below). I believe that addressing these concerns and the rest of my suggestions listed below can make this work stronger and can therefore change my score.

**Questions:**

### Additional motivation needed
- This work should draw clearer boundaries between pbRL and alternative reward learning methods and more generally alternative learning from human feedback methods. Right now, it is not very clear how and why this is different to IRL, for example, unless the reader is very familiar with [23]. This should not be the case, instead the paper should be really readable on its own and less dependent on prior work
    - Ideally, this should be done by proposing a stronger motivation for why pbRL (and more specifically meta-pbRL) as opposed to using something alternative (like IRL or meta-IRL).
    - It would be great to have a more exhaustive review of other meta-learning approaches from human feedback. A non-exhaustive list would include [i, ii, iii, iv], for example.

[i] Yu, L., Yu, T., Finn, C. and Ermon, S., 2019. Meta-inverse reinforcement learning with probabilistic context variables. NeurIPS 2019

[ii] Zhou, A., Jang, E., Kappler, D., Herzog, A., Khansari, M., Wohlhart, P., Bai, Y., Kalakrishnan, M., Levine, S. and Finn, C., 2019. Watch, try, learn: Meta-learning from demonstrations and reward. ICLR 2020

[iii] Finn, Chelsea, Tianhe Yu, Tianhao Zhang, Pieter Abbeel, and Sergey Levine. "One-shot visual imitation learning via meta-learning." CoRL 2017

[iv] Das, N., Bechtle, S., Davchev, T., Jayaraman, D., Rai, A. and Meier, F., 2020. Model-based inverse reinforcement learning from visual demonstrations. CoRL 2020

### Improving the Clarity of Expression
- The paper seems to refer to implicit conditioning on the predicted values from the Q function for the reward learning process as feedback but this is generally confusing since feedback has a very particular meaning in the context of robotics and robot learning. Specifically, **feedback refers to the mechanisms used for measuring the response over a specific variable** and this is not what is meant here. Similarly, the sentence '_[24] utilizes two kinds of feedback, initializes with imitation learning policy, and further improves the performance of the policy  with feedback from humans_' is confusing. Bootstrapping a policy does not mean the method uses demonstrations as feedback, similarly, using human feedback means actively correcting the system online, similarly to what Dagger-like methods do. Therefore, statements like this can be made a bit crisper. Similarly, other references to '_feedback_' in the paper seem equally confusing. An additional example is '_MRN outperforms prior methods in the case of little feedback_ ' - learning from a few labels does not mean there is a little feedback. I'd strongly encourage the authors to be clearer in the proposed expressions as to improve clarity.
- The authors mention that their '_intuition is that the optimization of the reward function needs both supervised learning and the feedback from the Q-function_' but the authors also conduct thorough experimentation ablating the use of those terms in appendix. Referring the readers to the relevant experiments would be great here too.
- It is overall great that the proposed work is doing better than the alternative solutions. However, it is not really clear why the performance of Preference PPO is so poor on the DMControl tasks. In particular, when comparing the reported results in Figure 3 with the original Preference-based paper [23] on those environments, it is really strange that Preference PPO does so poorly. It may be a good idea to clearly report the number of labels used to learn the rewards in the figures themselves. It took me a while to realise that there is a significant difference in the number of demos used in this work and the reported results in [23]. Adding a bit more discussion on this front can also be beneficial.
- Figure 4 is great, but it is really not clear what the upper-bound is. Therefore, it may be useful to add the achieved results with the ground truth reward using SAC. This should be easy, unless I am mistaken, those results already exist and are reported in Figures 2 and 3. Furthermore, it will be useful if the rows clearly indicated the names of the environments, similar to what is done in Figures 2 and 3.
- Conclusion summarises that the work is evaluated on robotic tasks but it should be more explicit, it is evaluated on robotic simulated tasks which is very different in practice.
- The paper also states that '_To our knowledge, we do not see any potential negative social influences of our work_' but the carbon footprint of the ran experiments and the required future runs for building on top of this (or any model-free approach) has a fairly negative impact. As a result, it may be better acknowledging this as opposed to stating that you do not see any negative social influences. Alternatively, just take out this sentence.
- The paper has a typo, namely '_we the mean squared error between_'' should be '_we use.._'

**Ethics Review Area:**

["I don’t know"]

**Limitations:**

This paper does not list the existing limitations of the proposed approach. I would really like to see a comprehensive list and a discussion of the existing limitations.

- A good example is that although it learns better from a few labels, it does not seem to solve the tasks with those few labels. This (along with the lacking discussion pro/con IRL) is a main reason for me to feel that the proposed solution has a fairly moderate impact. Perhaps a discussion on this topic would be great.
- The method also seems very sensitive to the number of meta updates.
- It also seems (from Fig. 2) that the learning process is not very stable and that some seeds completely failed to solve the tasks.
- Additionally, it is not clear how to scale this work to learning visual observations as it may require infeasible number of labels.

Discussing all those and potentially other limitations can be very useful.

**Strengths And Weaknesses:**

### Strengths
- A successful application of bi-level optimisation in the context of preference-based RL
- An overall thorough evaluation and nice ablations
- A fairly clear description of the algorithm

### Weaknesses
- lacks motivation for the use of this approach as opposed to alternatives (e.g. IRL)
- at times lacks clarity of expression

---

> ### Author Response · Authors · 2022-08-02
> **Response to Reviewer gArn (Part 2/2)**
>
> (Continue)
>
> **Q9: "Discussing all those and potentially other limitations can be very useful."**
>
> > **A9**: Thanks for your suggestions which makes good sense.
> >
> > 1\) As to the performance gap under few labels, we do not attempt to compete the performance of RL algorithms with ground-truth rewards. Instead, we want to achieve higher performance with fewer labels in the scenarios where preference labels are expensive to attain for PbRL. The difference to IRL is discussed in the answers to Q1, where obtaining human demonstrations can usually be more expensive than preference labels.
> >
> > 2\) As for the number of meta updates, it is related to the difficulty of the task. We investigate how this hyperparameter influences MRN in Appendix F. Figure 6 shows that MRN performs well when the number of meta updates is within a range, such as $[1000,7500]$ in Door Open task.
> >
> > 3\) As for the stability of learning curves, we find that the performance fluctuation of all methods is very large in Meta-world tasks. For example, even SAC algorithm has large performance fluctuations at Hammer and Sweep Into. In addition, we believe that few preference labels will bring more uncertainty, thus affecting the stability of algorithm performance.
> >
> > 4\) As for the visual observations, representation learning on visual input requires additional parameters and thus requires more data. While this is not the focus of this work, we consider this as future works.
> >
> > We will add more discussions of limitations in the revision.
>
> **Reference**
>
> -- [1] Paul F Christiano, Jan Leike, Tom Brown, Miljan Martic, Shane Legg, and Dario Amodei. Deep reinforcement learning from human preferences. In *Advances in Neural Information Processing Systems (NeurIPS)*, volume 30. Curran Associates, Inc., 2017.
>
> -- [2] Kimin Lee, Laura M Smith, and Pieter Abbeel. Pebble: Feedback-efficient interactive reinforcement learning via relabeling experience and unsupervised pre-training. In *International Conference on Machine Learning (ICML)*, volume 139 of *Proceedings of Machine Learning Research*, pages 6152–6163. PMLR, 18–24 Jul 2021.

---

> > ### Comment · Reviewer_gArn · 2022-08-06
> > **Thanks for your replies**
> >
> > I would like to thank the authors for the thorough and adequate response to my questions. **I am satisfied with the authors replies and I will therefore update my score to Accept.**
> >
> > In addition, I also read through some of the listed weaknesses and rebuttals of the other reviews. Overall, I find the updated manuscript and appendix very clear. Those now include a thorough theoretical analysis on the convergence properties and additional empirical evaluations.  Combined with the well prepared experimental set up and the presented novel methodology make this submission very suitable and valuable for the venue in my opinion.

---

> ### Author Response · Authors · 2022-08-02
> **Response to Reviewer gArn (Part 1/2)**
>
> We thank Reviewer gArn for your positive support and constructive comments. We provide our point-wise response below.
>
> **Q1: This work lacks motivation for the use of this approach as opposed to alternatives (e.g. IRL).**
>
> > **A1**: Thank you for your suggestion. 1) We want to point out one key difference between PbRL and IRL. IRL relies on human experts' demonstrations (usually high-quality) in completing specific tasks which are quite expensive and tries to inverse humans' reward functions. In PbRL, the human experts are only required to give preference labels on trajectory pairs generated by the training agent, which is much easier than directly providing demonstrations.
> >
> > 2\) We will review IRL, meta-IRL and give a clearer motivation of our work in the revision.
>
> ### **Response to questions in ''Improving the Clarity of Expression''**
>
> **Q2: Feedback refers to the mechanisms used for measuring the response over a specific variable.**
>
> > **A2**: We apologize for the confusion.
> >
> > 1\) The "feedback from the Q-function" means the implicit derivative of the outer loss based on the Q-function that facilitates the reward learning. We will revise the expression to make it clearer.
> >
> > 2\) The "feedback" in other places refers to human feedback, i.e., preference labels queried from human experts. We believe this meaning is consistent with your provided definition and also the PbRL literature. We will carefully revise our paper to avoid ambiguity.
>
> **Q3: "Referring the readers to the relevant experiments would be great."**
>
> > **A3**: Thank you for your suggestions. We will add proper refereces to link the experiments and claim in the paper.
>
> **Q4: "It is not really clear why the performance of Preference PPO is so poor on the DMControl tasks."**
>
> > **A4**: Thanks for your question. The performance of Preference PPO is highly influenced by the amount of preference labels available. In [1], the amount of feedback used in experiments is large. However, Preference PPO performs poorly in PEBBLE [2] with few preference labels. In our experimental setting, we further reduce the number, and Preference PPO performs worse than the results in PEBBLE. To reduce confusion, we will add the number of labels used for each task in  Figure 2 and 3 and more results discussion in the revision.
>
> **Q5: "Figure 4 is great, but it is really not clear what the upper-bound is."**
>
> > **A5**: We will add the results of SAC in Figure 4 and the name of the tasks in each row.
>
> **Q6: "Conclusion summarises that the work is evaluated on robotic tasks but it should be more explicit, it is evaluated on robotic simulated tasks which is very different in practice."**
>
> > **A6**: Thank you for your suggestions. We will revise to make it clearer.
>
> **Q7: Potential negative social impact.**
>
> > **A7**: Thanks, carbon footprint is surely a negative social impact. We will update it in the revision.
>
> **Q8: The paper has a typo.**
>
> > **A8**: Thank you for pointing out this. We will carefully revise any typos in the revision.
>
> (Citations included in Part 2)

---

> ### Author Response · Authors · 2022-08-02
> **A Reminder to Reviewer gArn**
>
> We have updated the manuscript and supplementary. The changes are highlighted in blue color.

---

### Official Review · Reviewer_hg7U · 2022-07-11

**Rating:** 7
**Confidence:** 4
**Soundness:** 3 good
**Presentation:** 3 good
**Contribution:** 3 good

**Summary:**

This paper proposes a novel algorithm called MRN for preference-based reinforcement learning, which models the preference order directly from the Q-value function instead of per-step rewards.

To realize this framework, this paper develops a meta-gradient-based method to learn reward functions from the supervision upon Q-values.

The experiment results show that MRN significantly outperforms baselines in terms of both sample efficiency and final policy quality.

**Questions:**

1. In section 5.3, what is the ground-truth Q-value? The policy preference is defined by binary comparison signals. How can the scale of $\hat r_\psi$ match the scale of unknown ground-truth rewards?

2. The introduction of bi-level optimization to preference-based RL is novel. However, the related work section can benefit from prior works on meta-gradient-based reward learning [1-2].

[1] Zhang et al. "On Learning Intrinsic Rewards for Policy Gradient Methods" NeurIPS 2018.

[2] Zhang et al. "What can learned intrinsic rewards capture?" ICML 2020.

**Limitations:**

No potential negative societal impact.

**Strengths And Weaknesses:**

The paper is well organized. The notations are clear and intuitive

The proposed algorithm is novel and the experiment results are solid.

---

> ### Author Response · Authors · 2022-08-02
> **Response to Reviewer hg7U**
>
> We thank Reviewer hg7U for your positive support. Below we provide point-wise responses to your questions.
>
> **Q1: "In section 5.3, what is the ground-truth Q-value? ... How can the scale of $\hat{r}_{\psi}$ match the scale of unknown ground-truth rewards?"**
>
> > **A1**: Thanks for your question.
> >
> > 1\) We did not perform Q-learning on ground-truth rewards, and "the ground-truth Q-value" in Section 5.3 means the empirical cumulative return based on 10 trajectories collected.
> >
> > 2\) We did not attempt to match the scale of $\hat{r}_{\psi}$ to the ground-truth rewards, nor is it known. The implementation of $\hat{r}_\psi$ utilizes a tanh function to convert the reward into $(-1, 1)$, which is also a widely adopted trick in earlier works [1][2].
> >
> > 3\) To reasonably evaluate the accuracy of learned Q-function, empirical cumulative returns from the ground-truth rewards are normalized to match the scale of $\hat{r}_{\psi}$. The results show that MRN performs better approximation to the ground-truth rewards at the same scale.
>
> **Q2: The related work section can benefit from prior works on meta-gradient-based reward learning.**
>
> > **A2**: Thank you for pointing out these relevant works. We will discuss these works in the revised draft.
>
> **Reference**
>
> -- [1] Kimin Lee, Laura M Smith, and Pieter Abbeel. Pebble: Feedback-efficient interactive reinforcement learning via relabeling experience and unsupervised pre-training. In *International Conference on Machine Learning (ICML)*, volume 139 of *Proceedings of Machine Learning Research*, pages 6152–6163. PMLR, 18–24 Jul 2021.
>
> -- [2] Jongjin Park, Younggyo Seo, Jinwoo Shin, Honglak Lee, Pieter Abbeel, and Kimin Lee. SURF: Semi-supervised reward learning with data augmentation for feedback-efficient preference-based reinforcement learning. In *International Conference on Learning Representations (ICLR)*, 2022.

---

> ### Author Response · Authors · 2022-08-03
> **A Reminder to Reviewer hg7U**
>
> We have updated the manuscript and supplementary. The changes are highlighted in blue color.

---

### Official Review · Reviewer_5HHg · 2022-07-11

**Rating:** 4
**Confidence:** 4
**Soundness:** 2 fair
**Presentation:** 3 good
**Contribution:** 2 fair

**Summary:**

This paper propose MRN, a feedback-efficient PbRL framework that incorporates bi-level optimization for both reward and policy learning. The key idea is to incorporate a new auxiliary task into standard PbRL framework, through utilizing the performance of the Q-function to predict preference label. Multiple experiments are conduct to verify MRN's superiority over various PbRL baselines in terms of high feedback efficiency and cumulative return.

**Questions:**

（1）Is it reasonable to utilize $Q_{\theta}(s_{0}^{0}, a_{0}^{0})$ to represent the return of the segment $\sigma^{0}$ in Eq. (6)? $Q_{\theta}(s_{0}^{0}, a_{0}^{0})$ represents the cumulative return of continuing executing $\pi$ after selecting action $a_0$ at state $s_0$. However, the behavior performed by $\pi$ may not be consistent with segment $\sigma^{0}$ at subsequent state $s_{t}^{0}, where $t>0$.
(2) Although the experiment is relatively sufficient, there is still a lack of rational explanation, why using the Q value to give the pseudo-label will bring about the high feedback efficiency?
(3) Why MRN performs far less than SAC on almost all experiments？
(4) Why does MRN perform only marginally better than other baselines in the DMControl environment?

**Limitations:**

This paper lacks to address the limitations of their work. （1）Since reward learning and policy learning are performed iteratively, it is not clear whether the convergence of the algorithm can be guaranteed. （2）The extent to which the performance of reward learning depends on the quality of human feedback.

**Strengths And Weaknesses:**

Originality:
Although MRN is the first for PbRL, its key ideas have been proposed and widely used in computer vision.

Quality：
The central idea of this paper is straightforward. I have two key doubts as follows。（1）Is it reasonable to utilize $Q_{\theta}(s_{0}^{0}, a_{0}^{0})$ to represent the return of the segment $\sigma^{0}$ in Eq. (6)? $Q_{\theta}(s_{0}^{0}, a_{0}^{0})$ represents the cumulative return of continuing executing $\pi$ after selecting action $a_0^0$ at state $s_0^0$. However, the behavior performed by $\pi$ may not be consistent with segment $\sigma^{0}$ at subsequent state $s_{t}^{0}$, where $t>0$. (2) Although the experiment is relatively sufficient, there is still a lack of rational explanation, why using the Q value to give the pseudo-label will bring about the high feedback efficiency?

Clarity:
The main content is clear, but there is a lack of rational explanation for some key points. For example, it is unclear why using the Q value to give the pseudo-label will bring about the high feedback efficiency?

Significance:
The work is significant for improving feedback efficiency in PbRL, but I doubt the soundness of the main idea.

---

> ### Author Response · Authors · 2022-08-02
> **Response to Reviewer 5HHg**
>
> We thank Reviewer 5HHg for the constructive comments and provide our point-wise response below.
>
> **Q1: "Is it reasonable to utilize $Q_{\theta}(s_0^0, a_0^0)$ to represent the return of the segment $\sigma^0$ in Eq. (6)?"**
>
> > **A1**: Thanks for your question. 1\) Compared to earlier works that use total reward, we propose to use the Q-value for segment evaluation. MRN uses the Q-value to evaluate segments which can reduce the variance of estimation compared to previous methods using the return of segments. 2\) The mismatch between the behavior policy and current policy $\pi$ is expected, as Q-function is updated based on off-policy learning, and it is later used to criticize policy $\pi$, which is widely adopted in off-policy RL algorithms.
>
> **Q2: "Although the experiment ..., why using the Q value to give the pseudo-label will bring about the high feedback efficiency?"**
>
> > **A2**: Thanks for your question. 1\) Previous works only align the reward function with human preferences. However, aligning the Q-value with preference labels in MRN is a more direct way to learn a well-performed policy because the policy directly learns from the Q-value.
>
> **Q3: "Why MRN performs far less than SAC on almost all experiments?"**
>
> > **A3**: We apologize for the confusion. SAC is evaluated as the upper bound as it used the ground-truth rewards instead of preference labels. MRN and other PbRL methods have no access to the ground-truth rewards but human preference labels. We will revise Figure 2 and 3 to clarify this in the updated paper.
>
> **Q4: "Why does MRN perform only marginally better than other baselines in the DMControl environment?"**
>
> > **A4**: Thanks for your question. 1\) The metrics in the two sets of tasks are different and are not straightforwardly comparable. Meta-world uses success rate in completing tasks, and DMControl uses the cumulative return in learning specific behaviours. 2\) In Figure 3, though the learning curves seems close to each other, MRN outperforms PEBBLE by 22.4\% in Walker, 18.3\% in Cheetah, and 24.7\% in Quadruped in terms of the cumulative return.
>
> **Q5: "Since reward learning and policy learning are performed iteratively, it is not clear whether the convergence of the algorithm can be guaranteed."**
>
> > **A5**: Thanks for pointing our this. We have examined our algorithm and we can show that our method converges to the critical points of both the outer loss (preference prediction loss based on the Q-function) and the inner loss (Q-learning loss) under some mild conditions. We will add the theoretical results in Section 4.3 in the revision.
>
> **Q6: "The extent to which the performance of reward learning depends on the quality of human feedback."**
>
> > **A6**: Thanks for your question. While this is not the focus of this work, we have shown in Figure 8 (Appendix F) that MRN is robust to 10\% noisy preference labels.
> >
> > To understand the boundaries of MRN, we conduct experiments on Walker and Door Open tasks with ratios of noisy data as {0%, 10%, 15%, 20%, 25%, 30%}. The results in the following tables show that MRN fails with 15\% noisy data in Walker and fails with 30\% noisy data in Door Open.
> >
> >
> > Table 1: Performance of MRN with different noise levels
> >
> > | Env/Noise | 0%   | 10%  | 15%     | 20%  | 25%  | 30%    |
> > | --------- | ---- | ---- | ------- | ---- | ---- | ------ |
> > | Walker    | 710  | 604  | **305** | /    | /    | /      |
> > | Door Open | 80%  | 66%  | 66%     | 66%  | 66%  | **0%** |
> >
> > Table 2: Performance of PEBBLE with different noise levels
> >
> > | Env/Noise | 0%   | 10%     | 15%     | 20%  | 25%  |
> > | --------- | ---- | ------- | ------- | ---- | ---- |
> > | Walker    | 594  | **410** | /       | /    | /    |
> > | Door Open | 68%  | 50%     | **44%** | /    | /    |
> >
> > Table 3: Performance of SURF with different noise levels
> >
> > | Env/Noise | 0%   | 10%     | 15%  | 20%     | 25%  |
> > | --------- | ---- | ------- | ---- | ------- | ---- |
> > | Walker    | 605  | 525     | 522  | **371** | /    |
> > | Door Open | 60%  | **37%** | /    | /       | /    |

---

> > ### Comment · Reviewer_5HHg · 2022-08-06
> > **The reviewer‘s response**
> >
> > I remain skeptical about your response to Q1 and Q2. Although Q-value may reduce evaluation variance, it may also introduce evaluation bias. The current practice is still heuristic, and there is a lack of plausible evidence (eg, theoretical analysis) to support the ideas in this paper.

---

> > > ### Author Response · Authors · 2022-08-06
> > > **Response to Reviewer 5HHg**
> > >
> > > We thank Reviewer 5HHg for the response and provide our point-wise response below. Kindly let us know if you have further concerns. we are happy to address any of your questions.
> > >
> > > **Q1: "Although Q-value may reduce evaluation variance, it may also introduce evaluation bias."**
> > >
> > > > **A1**: 1\) We conduct experiments to evaluate the mean squared error between ground-truth Q-value and the predicted Q-value. The results in Table 1 show that MRN learns a more accurate Q-function than other PbRL methods, indicating that the bias of the predicted Q-value is small.
> > > >
> > > > 2\) Bradley-Terry model is bias-friendly according to its properties [1][2]. On the other hand, using Q-value for segment evaluation can reduce variance of estimation. Therefore, it is reasonable to replace total reward with the Q-value in segment evaluation.
> > >
> > > **Q2: "The current practice is still heuristic."**
> > >
> > > > **A2**: MRN reduces the confirmation bias, which is a problem that a network overfits to inaccurate targets predicted by another network [3]. When there are few preference labels, previous PbRL methods, such as PEBBLE, SURF and RUNE will likely learn an inaccurate reward function, leading to an inaccurate Q-function. MRN reduces the confirmation bias since reward learning is aware of how the predicted rewards influences the performance of the Q-function. Thus, when the reward function is not accurate caused by insufficient preference labels, MRN can lower the affects of inaccurate predicted rewards on the Q-function. And the experimental results in Table 1 demonstrate that MRN learns a more accurate Q-function than the baselines.
> > >
> > > **Reference**
> > >
> > > -- [1] Ralph Allan Bradley and Milton E. Terry. Rank analysis of incomplete block designs: I. the method of paired comparisons. Biometrika, 39(3/4):324–345, 1952.
> > >
> > > -- [2] Kenneth J Koehler and Harold Ridpath. An application of a biased version of the bradley-terry-luce model to professional basketball results. *Journal of mathematical psychology*, 25(3):187–205, 1982.
> > >
> > > -- [3] Antti Tarvainen and Harri Valpola. Mean teachers are better role models: Weight-averaged consistency targets improve semi-supervised deep learning results. In *Advances in Neural Information Processing Systems*, volume 30. Curran Associates, Inc., 2017.

---

> > > > ### Author Response · Authors · 2022-08-08
> > > > **Response to Reviewer 5HHg**
> > > >
> > > > Dear Reviewer 5HHg,
> > > >
> > > > We would like to know whether we have addressed your concerns. If so, might you be able to update your rating to reveal this? Otherwise, please let us know whether you have further questions. We are happy to answer that.
> > > >
> > > > Authors

---

> > > > > ### Comment · Reviewer_5HHg · 2022-08-09
> > > > > **Feedback**
> > > > >
> > > > > I still doubt the soundness of the main idea. Although the experiment is relatively sufficient, there is still a lack of rational explanations for Q1 and Q2. I insist on this rating.

---

> > > > > > ### Comment · Reviewer_gArn · 2022-08-09
> > > > > > **It is unclear what 5HHg requires**
> > > > > >
> > > > > > I would first like to highlight the value of all reviews and the utility of discussing. In this context I would like to intervene this discussion.
> > > > > >
> > > > > >  Specifically, I cannot understand what does reviewer 5HHg wants to see in support of Q1 and Q2 and why.
> > > > > >
> > > > > > The authors have included a thorough evaluation and a comprehensive reference to the valid literature in support of their claims with respect to the reviewer’s questions. In addition, they provide a very thorough convergence analysis, a segment that typically lacks in most empirically structured related works.
> > > > > >
> > > > > > In light of this and in the name of constructive criticism, I would like to invite reviewer 5HHg to list a specific and comprehensive list of requirements that will clear out the confusion with respect to Q1 and Q2 and why they do not accept the authors rebuttal. Right now I, as a reviewer, am a bit confused and I do not feel comfortable in the presence of ambiguity. Thank you for participating in an enlightening discussion.

---

> ### Author Response · Authors · 2022-08-03
> **A Reminder to Reviewer 5HHg**
>
> We have updated the manuscript and supplementary. The changes are highlighted in blue color.

---

### Official Review · Reviewer_tWWp · 2022-07-11

**Rating:** 7
**Confidence:** 4
**Soundness:** 4 excellent
**Presentation:** 4 excellent
**Contribution:** 4 excellent

**Summary:**

The paper introduces the meta-reward-net (MRN) method for preferenced-based reinforcement learning (PbRL). MRN learns a reward from human preferences over trajectory pairs by combining a supervised preference-based loss with a bi-level optimization loss that optimizes the reward based on the Q-function's ability to reflect human preferences. Experiments show that MRN outperforms state-of-the-art baselines in PbRL. Furthermore, MRN improves the efficiency of using feedback labels. Lastly, the paper analyzes the source MRN's gains as coming from a more accurate Q-function.

**Questions:**

* The paper only considers a single pseudo update of the Q-function when computing the outer-loop loss. Did the work consider multiple psuedo updates? This could result in greater efficiency due to more accurate meta-gradient updates.
* What is the impact of the weighting of the two losses in MRN? In MRN, is the bi-level loss and the auxiliarly loss weighted the same? While, Section D analyzes different update frequencies, is the magnitude of these losses also important?

**Limitations:**

The paper analyzes the limitations of MRN through the feedback comparison experiments in Section 5.3. These results show that MRN is limited to working best relative to baselines when there is little available avaiable feedback. However, a more explicit discussion of the limitations of the approach would be valuable.

**Strengths And Weaknesses:**

Strengths:
* The methodology is novel and well motived for the problem. Updating the reward based on the Q-function's performance is a sensible way to provide more reward supervision as this will update the reward to be better suited for the task.
* MRN has the novel insight of introducing bi-level optimization into PbRL. Bi-level optimization for PbRL could open future research into using other metrics in the outer loss.
* MRN consistently outperforms state-of-the-art prior work in all the considered benchmarks. Furthermore, the benchmarks are standard in the field and a fair assessment of MRN and baselines.
* Figure 4 in Section 5.3 demonstrates that MRN is preferred over baselines when less feedback is available.
* Analysis of the Q-function accuracy in Section 5.3 helps analyze the source of performance gains. MRN learns a more accurate Q-function, which enables MRN to perform better with less feedback than baselines.
* The paper is well written and clear.

Weaknesses:
* Can MRN outperform baselines even with more feedback? In Figure 4 h,d, where methods receive 2k feedback labels, all methods perform similarly. Or can MRN help PbRL scale to more complex tasks better than baselines, even with more feedback? Since the amount of feedback greatly impacts performance, how was the amount of feedback selected for the experiments in Figures 2 and 3?
* There is still a large gap between MRN and the performance of the expert with the ground truth reward. What limitations of MRN, and PbRL in general, prevent it from reaching the performance of the expert?

---

> ### Author Response · Authors · 2022-08-02
> **Response to Reviewer tWWp**
>
> We thank Reviewer tWWp for your positive support. Below we provide point-wise response to your questions.
>
> **Q1: "Can MRN outperform baselines even with more feedback? ... Or can MRN help PbRL scale to more complex tasks better than baselines, even with more feedback?"**
>
> > **A1**: Yes, Figure 2 shows that MRN outperforms baselines on complex tasks such as Sweep Into (feedback=4000) and Hammer (feedback=10000).
> >
> > From Figure 4, all PbRL methods perform well when enough feedback is provided, but MRN performs well with a smaller amount of feedback, evidencing that MRL is more feedback efficient.
>
> **Q2: "Since the amount of feedback ..., how was the amount of feedback selected for the experiments in Figures 2 and 3?"**
>
> > **A2**: We have reported the amount of feedback in Section 5.1. *For the amount of human’s preference feedback, we set 100 for Walker, Cheetah, Button Press and Window Open, 700 for Quadruped, 1000 for Door Open and Drawer Open, 4000 for Sweep Into, and 10000 for Hammer.* To make this clearer, we will add the number of feedback in the title of Figure 2 and 3 in the revision.
>
> **Q3: "What limitations of MRN, and PbRL in general, prevent it from reaching the performance of the expert?"**
>
> > **A3**: Thanks for your question. One limitation  shared by MRN and other PbRL methods is that, the preference can only reflect the quality of trajectories; and the fine grained preference on specific actions is not available.
> >
> > However, we argue that the performance of MRN matches the performance of SAC using the ground-truth reward in tasks such as Walker and Door open, as shown in Figure 4 with sufficient human feedback. We will add SAC as the upper bound in Figure 4 in the revision.
>
> **Q4: "Did the work consider multiple psuedo updates?"**
>
> > **A4**: Thanks for your question. In our experiments, we consider single pseudo updates because this version performs well enough with less computation cost than the multi-step version. To show the performance under multiple pseudo updates, we conducted experiments with different pseudo updating steps including: {$1, 5, 10, 15$} and the results are shown in the following table.
> >
> > | Env/Pseudo Updating Step(s) |  $1$  |  $5$  | $10$  | $15$  |
> > | :-: | :-: | :-: | :-: | :-: |
> > |           Walker            | $772$ | $823$ | $652$ | $486$ |
> >
> > It shows that multi-step pseudo updating have positive effects with $5$ steps pseudo updating. When the number of steps becomes larger, the performance of MRN becomes worse. We hypothesize that this is due to the output of the reward function is inaccurate at the initial stage. If MRN pseudo updates too many times, the copy of the Q-function will be led to an incorrect direction by the inaccurate reward estimation. Therefore, the implicit derivative calculated based on the copied Q-function is not accurate. With this process going on, the reward function and the Q-function move far from the right direction together.
> >
> > Although MRN benefits from multi-step pseudo updating, more pseudo updating steps bring about more computation cost. In our implementation, we just use the single pseudo updating version.
>
> **Q5: "What is the impact of the weighting of the two losses in MRN?"**
>
> > **A5**: Thanks for your question. The weighting of auxiliary supervised loss is set to $1$ in our experiments and report promising performance. We additionally conduct experiments of different weights. We set the weight of the outer loss to $1$ and the weight of supervised loss is set to {$0, 0.01, 0.1, 1, 10, 100$}, respectively. The results averaged across three runs are shown in the following table.
> >
> > | Env/Weight | $0$  | $0.01$ | $0.1$ |  $1$  | $10$  | $100$ |
> > | :--: | :--: | :---: | :--: | :--: | :--: | :--: |
> > |   Walker   | $44$ | $327$  | $476$ | $710$ | $390$ | $522$ |
> >
> > As the weight goes up, the performance of MRN increases first then decreases, reaching the highest point when the two losses weighted the same. MRN works poorly when the weight of supervised loss is $0$. We hypothesize this is because reward learning is only aware of the performance of the Q-function, but ignores the ground-truth preference labels. And as the weight of supervised loss increases, the performance of MRN is gradually close to the performance of PEBBLE [1].
>
> **Q6: "A more explicit discussion of the limitations of the approach would be valuable."**
>
> > **A6**: Thanks for your question. The possible limitations of MRN include: 1) MRN relies on qualities of human feedback; 2) MRN cannot discriminate between good and bad actions within one trajectory. We will add this discussion in the revision.
>
> **Reference**
>
> -- [1] Kimin Lee, Laura M Smith, and Pieter Abbeel. Pebble: Feedback-efficient interactive reinforcement learning via relabeling experience and unsupervised pre-training. In *International Conference on Machine Learning (ICML)*, volume 139 of *Proceedings of Machine Learning Research*, pages 6152–6163. PMLR, 18–24 Jul 2021.

---

> ### Author Response · Authors · 2022-08-03
> **A Reminder to Reviewer tWWp**
>
> We have updated the manuscript and supplementary. The changes are highlighted in blue color.

---

### Meta-Review · Area_Chair_gu3m · 2022-08-26

**Recommendation:** Accept
**Confidence:** Less certain

**Metareview:**

The reviewers carefully analyzed this work and agreed that the topics investigated in this paper are important and relevant to the field. Overall, the reviewers had a generally positive impression of this paper. One reviewer argued that the paper introduces a novel technique for learning a reward function from human preferences. The reviewer acknowledged that the method outperforms SOTA baselines, that it is well-motivated, and that it introduces important insights. As the main weakness, this reviewer pointed out that the paper analyses of the method's limitations show that it works best w.r.t. baselines when there is little available feedback. They encourage the authors to include a more explicit discussion on limitations such as this. Another reviewer had a less favorable view of this work and argued that the key ideas in this paper have been proposed and widely used in computer vision. They had two main technical questions, to which the authors responded. The reviewer, however, remained concerned about whether, e.g., using Q-values could introduce bias and if the method lacked sufficient theoretical analyses. Two other reviewers had more positive views of this paper. One of them argued that the paper introduced a novel algorithm and that its experiments showed that it outperforms baselines both in sample efficiency and final policy quality. The authors responded to the technical questions made by this reviewer, and the reviewer said they were satisfied with the authors' rebuttal. Finally, a fourth reviewer also acknowledged that this method is novel and that it was thoroughly evaluated in simulation and compared to reasonable baselines, where it was shown to be faster/more efficient. This reviewer initially thought there was insufficient motivation for using this approach over alternative meta-learning techniques—in which case the paper's impact could be moderate. However, the authors addressed the reviewer's concerns/questions in detail and the reviewer seemed satisfied with the rebuttal, changing their score to Accept. Overall, thus, it seems like most reviewers were positively impressed with the quality of this work. They look forward to an updated version of the paper that addresses the suggestions mentioned in their reviews and during the discussion phase.

**Award:**

No

---

### Decision · Program_Chairs · 2022-09-14

Accept